# Fine-tuning is Not Enough: Rethinking Evaluation in Molecular Self-Supervised Learning

## Abstract

Self-Supervised Learning (SSL) has shown great success in language and vision by using pretext tasks to learn representations without manual labels. Motivated by this, SSL has also emerged as a promising methodology in the molecular domain, which has unique challenges such as high sensitivity to subtle structural changes and scaffold splits, thereby requiring strong generalization ability. However, existing SSL-based approaches have been predominantly evaluated by naïve fine-tuning performance. For a more diagnostic analysis of generalizability beyond fine-tuning, we introduce a multi-perspective evaluation framework for molecular SSL under a unified experimental setting, varying only the pretraining strategies. We assess the quality of learned representations via linear probing on frozen encoders, measure Pretrain Gain by comparison against random initialization, quantify forgetting during fine-tuning, and explore scalability. Experimental results show that several models, surprisingly, exhibit low or even negative Pretrain Gain in linear probing. Graph neural network-based models experience substantial parameter shifts, and most models derive negligible benefits from larger pretraining datasets. Our reassessments offer new insights into the current landscape and challenges of molecular SSL.

## 1 Introduction

Recently, Self-Supervised Learning (SSL) has achieved significant success in natural language processing (NLP) Devlin et al. (2019); Floridi & Chiriatti (2020) and computer vision (CV) Dosovitskiy et al. (2020); Grill et al. (2020); He et al. (2022). SSL has received growing attention to learn useful representations from large-scale unlabeled data Chen et al. (2020); Radford et al. (2021). Motivated by this success, SSL has also emerged as a promising approach in the molecular domain Li & Jiang (2021); Moon et al. (2023); Son et al. (2025), where labeling molecular data is expensive and time-consuming because it relies on real-world experiments Juan et al. (2024); Wouters et al. (2020).

The molecular field presents several unique challenges for designing generalizable models. For instance, downstream tasks in this domain are diverse, predicting toxicity, solubility, and estimating bioactivity Lipinski et al. (1997). In addition, molecular properties are often highly sensitive to even subtle structural changes; a small modification in an atom or bond can lead to significant differences in biological activity or chemical property Kubinyi (2002). When evaluating such properties in downstream tasks, model generalization is commonly assessed using random splits. However, in the molecular domain, scaffold splitting is used, due to molecules with similar core structures tend to have similar properties Bemis & Murcko (1996). Scaffold splitting ensures that the test set contains core structures unseen during training.

To solve these challenges of the molecular domain, various molecular SSL have been proposed. However, as shown in Table 1, existing molecular SSL have primarily been evaluated by naïve fine-tuning performance. This evaluation may not be sufficient for thoroughly assessing the generalizability of pretrained representations, as fine–tuning modifies all parameters and can thereby lead to forgetting of knowledge acquired during large–scale pretraining Zhou & Cao (2021). Moreover, fair comparisons have not been conducted, as each study employs different downstream prediction heads, hidden dimensions, and dataset scales. For example, downstream prediction heads range from one-layer Hu et al. (2019); Xu et al. (2021) to two-layer MLPs Rong et al. (2020); Fang et al. (2023); hidden dimensions vary from 300 Hu et al. (2019); Sun et al. (2022) to 1200 Rong et al. (2020); and

Table 1: Summary of existing molecular SSL methods. Evaluation indicates which metric was used to evaluate each model.Experimental Configuration describes the pretraining dataset size and model architecture used for each method.

| Model | Evaluation | | | | Experimental Configuration | | | |
|-------|-----------|--------|------|--------------|---------------|----------|------------|-------------|
| | Fine-tune | Random | Gain | Data Scaling | Pretrain Data | Backbone | Hidden Dim | # Parameter |
| GROVER Rong et al. (2020) | ✓ | ✓ | | ✓ | 11.00 M | Transformer | 1200 | 5,418K |
| AttributeMask Hu et al. (2019) | ✓ | ✓ | ✓ | | 2.00 M | GNN | 300 | 1,857K |
| ContextPred Hu et al. (2019) | ✓ | ✓ | ✓ | | 2.00 M | GNN | 300 | 1,857K |
| EdgePred Hamilton et al. (2017) | ✓ | ✓ | ✓ | | 2.00 M | GNN | 300 | 1,857K |
| GraphLoG Xu et al. (2021) | ✓ | ✓ | | | 2.00 M | GNN | 300 | 1,857K |
| GraphCL You et al. (2020) | ✓ | ✓ | | | 2.00 M | GNN | 300 | 1,857K |
| KANO Fang et al. (2023) | ✓ | | | | 0.25 M | GNN | 300 | 2,088K |
| ChemBERTa Chithrananda et al. (2020) | ✓ | | | ✓ | 77.00 M | Transformer | 768 | 3,683K |

pretraining data sizes span from 0.25 million Fang et al. (2023) to 77 million samples Chithrananda et al. (2020). These highlight the need for a multi-perspective and fair evaluation strategy.

To systematically analyze molecular SSL beyond fine-tuning, we propose a multi-perspective evaluation framework for molecular SSL. Since prior studies have been evaluated under different experimental configurations as shown in Table 1, it hinders fair comparisons regarding the effectiveness of pretraining. All non-pretraining factors — such as datasets, prediction heads, and hidden dimensions — are kept the same, while only pretraining-related configurations are varied. Upon this unified setup, we propose various evaluation metrics to assess molecular SSL. We utilize linear probing to evaluate the quality of pretrained representations. We introduce the Pretrain Gain to measure the benefits of pretraining against random initialization. We quantify forgetting during fine-tuning through parameter shifts. Finally, we explore the scalability to evaluate their potential as foundation models. These metrics allow us to reassess existing approaches and provide insights into the generalization of pretrained representations in molecular SSL.

Our contributions are summarized as follows:

- A unified experimental setup is employed that standardizes experimental variables (e.g., hidden dimensions, downstream heads, and datasets) across diverse molecular SSL methods, enabling fair and controlled comparisons focused solely on pretraining strategies.

- We propose a multi-perspective evaluation framework for molecular SSL beyond fine-tuning. It includes linear probing to assess representation quality, Pretrain Gain to quantify pretraining benefits, parameter shift analysis to measure forgetting, and scalability.

- Comprehensive reassessments offer new insights into the current landscape and challenges of molecular SSL, revealing that, surprisingly, several models exhibit low or even negative Pretrain Gain, substantial parameter shifts, and negligible benefits from increased scale.

## 2 PRELIMINARIES

### 2.1 SELF-SUPERVISED LEARNING

SSL leverages unlabeled data to reduce reliance on manual annotation Devlin et al. (2019); Radford et al. (2021); Kingma et al. (2019). It typically follows a two-stage framework: pretraining and downstream. Pretraining learns generalizable representations by capturing intrinsic patterns within large-scale unlabeled datasets Tendle & Hasan (2021); Goyal et al. (2019); Fang et al. (2024). These results suggest that these generalized representations enable efficient transfer to downstream tasks with limited labeled data. The downstream step connects a task-specific prediction head to the pretrained encoder. The transferred model is then trained with labeled data to perform the target task. These tasks include toxicity prediction, solubility estimation, binding affinity prediction, and other molecular property classification or regression tasks.

### 2.2 PRETRAINING STRATEGIES AND ARCHITECTURES FOR MOLECULAR SSL

To understand molecular SSL, we organize existing approaches by categorizing pretext tasks into four types — generation-based, auxiliary property–based, contrast-based, and hybrid — and by analyzing model architectures, focusing on GNN-based and Transformer-based designs Liu et al. (2022); Xu et al. (2018); Rong et al. (2020); Chithrananda et al. (2020).

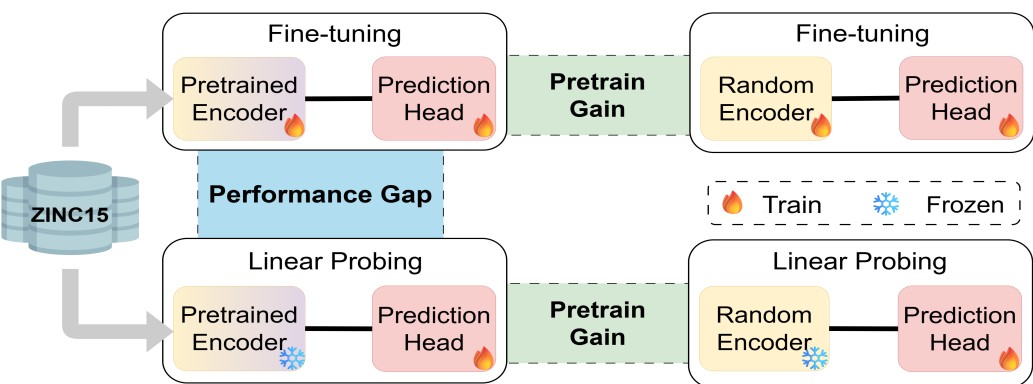

Figure 1: The left part presents results using a pretrained encoder in fine-tuning and linear probing, while the right part shows the same experiments with a randomly initialized encoder. To quantify the benefit of pretraining, we compare models under identical training settings except for the encoder. We assess the generality of the learned representations by comparing fine-tuning and linear probing: high performance under linear probing suggests general representations.

Generation-based methods Hou et al. (2022); Wang et al. (2019) define the pretext task as reconstructing masked components of molecular data, such as atom types, bond types, or substructures. For example, certain atoms or bonds in a molecular graph, or tokens in string-based SMILES Weininger (1988); Krenn et al. (2022; 2020), are masked during pretraining, and the model is trained to recover them. In our study, AttributeMask Hu et al. (2019), EdgePred Hamilton et al. (2017), and ChemBERTa Chithrananda et al. (2020) are classified as Generation-based methods. Auxiliary property-based methods Zhang et al. (2021); Hu et al. (2019) utilize inherent chemical or structural properties of molecules, such as atom degrees, aromaticity, and Motif Zhang et al. (2021), as a prediction target. The ContextPred Hu et al. (2019) model is an example of this approach. Contrast-based methods You et al. (2021) learn representations by contrasting augmented views of molecules, typically generated through atom, edge, and subgraph level perturbations. The model learns to make representations of views from the same molecule similar, while making those from different molecules dissimilar. GraphLoG Xu et al. (2021), GraphCL You et al. (2020), and KANO Fang et al. (2023) are included in this category. Hybrid methods Zang et al. (2023) combine several pretext tasks to capture more complex structures. For example, GROVER Rong et al. (2020) learns a pretext task that combines generation-based objectives with auxiliary property prediction.

Molecular SSL commonly employs two main model architectures: GNN and Transformer. GNNs are particularly effective at capturing the structural properties of molecular graphs, in which atoms are represented as nodes and chemical bonds as edges. Schütt et al. (2018); Scarselli et al. (2008). Through message passing, nodes iteratively aggregate information from their neighbors, enabling the model to capture the underlying graph structure Gilmer et al. (2017). This allows GNNs to learn representations that include both atomic-level information and global structural context. Transformer-based models commonly use sequence-based inputs, such as SMILES Li & Jiang (2021); Chithrananda et al. (2020); Wang et al. (2019). Unlike GNNs, these models do not require an explicit graph structure and instead learn relational patterns from sequential data. As a hybrid, GROVER incorporates GNNs and Transformer-style attention to node features instead of using sequence-based inputs. GNNs are used to extract graph structure

## 2.3 PRETEXT TASK OF MOLECULAR SELF-SUPERVISED LEARNING

We provide a summary of the pretext tasks used in the existing molecular SSL methods employed in our experiments.

- **GROVER** is a hybrid model that learns a pretext task using both subgraph masking and motif prediction. Subgraph masking aims to reconstruct masked substructures, while motif prediction is RDKit-extracted chemical motifs for multi-label classification Landrum et al. (2013).

- **AttributeMask** predicts masked properties of nodes.
- **ContextPred** predicts whether a neighborhood graph and a context graph belong to the same node. It learns through a classification task with negative sampling.
- **EdgePred** predicts the adjacency matrix of a graph
- **GraphLoG** uses a hierarchical prototype structure via clustering, enabling contrastive learning between local instances and their parent prototypes.
- **GraphCL** is a contrastive learning by generating augmented graph views through node and edge masking.
- **KANO** is contrastive learning between original and augmented graphs, where augmentation is performed by adding atomic information from a knowledge graph. In addition, a prompt approach is used to bridge the gap between pretraining and the downstream task
- **ChemBERTa** predicts masked tokens in SMILES strings.

## 3 Multi-perspective Evaluation Framework for Molecular SSL

We design various evaluation strategies for a more systematic and diagnostic generalization analysis beyond fine-tuning, an overview is shown in Figure 1.

### 3.1 Quality of Learned Representations via Linear Probing

In molecular SSL, pretrained models are mainly evaluated by fine-tuning. However, since fine-tuning updates all parameters of both the encoder and the prediction head, there is a risk that the pretrained representations may be significantly changed. This makes it hard to distinguish whether the improved performance is due to the quality of the pretrained representations or the encoder being changed by downstream data during fine-tuning.

To separate these effects and focus the evaluation on the quality of pretrained representations, we employ linear probing, the encoder is frozen to preserve its pretrained representations, and trains only the prediction head. This allows us to evaluate the focus on the quality of the pretrained representations, and high performance in linear probing indicates that the representations are generalized. However, the quality of pretrained representations has rarely been evaluated using linear probing in previous molecular SSL studies.

### 3.2 Pretrain Gain Against Random Initialization

We introduce Pretrain Gain, a metric for quantitatively measuring the performance improvement achieved through pretraining. It is computed by comparing the performance of a model using pretrained parameters and randomly initialized parameters. Specifically, under the same model architecture and training settings, only the encoder parameter differs: one uses pretrained weights, while the other is randomly initialized. Since only the parameters differ in this setup, the performance difference can be regarded as the effect of pretraining. The formula is as follows:

$$\text{Pretrain Gain} = \frac{\text{Score}_{\text{pretrain}} - \text{Score}_{\text{random}}}{\text{Score}_{\text{random}}} \times 100 \tag{1}$$

Here, $\text{Score}_{\text{pretrain}}$ and $\text{Score}_{\text{random}}$ denote the downstream performance of models using pretrained and randomly initialized encoders, respectively. By dividing by $\text{Score}_{\text{random}}$, the formula calculates the relative improvement over the $\text{Score}_{\text{random}}$ baseline as a ratio, which is then converted into a percentage.

### 3.3 Quantifying Forgetting through Parameter Shift

fine-tuning updates all model parameters, and thus, the pretrained encoder may also be modified. As a result, the pretrained knowledge can be partially or completely forgotten during fine-tuning. This issue can be mitigated when the pretrained representations are sufficiently general, allowing the encoder to align across various tasks with minimal changes. In contrast, when the representations lack

generality, the encoder requires substantial modification to align with the downstream task Zhang et al. (2020).

To investigate forgetting, we quantitatively measure the parameter shift during fine-tuning. The parameter shift is computed as the L2 distance between the pretrained encoder parameters before and after fine-tuning. It is calculated as:

$$\Delta_{\text{param}} = \sum_{i=1}^{N} \left\| \theta_i^{\text{before}} - \theta_i^{\text{after}} \right\|^2 \tag{2}$$

Here, $\theta_{\text{before}}$ and $\theta_{\text{after}}$ denote the encoder parameters before and after fine-tuning. By comparing the two, we aim to quantify the extent of the parameter shift. A larger value of $\Delta_{\text{param}}$ indicates that the encoder parameters have significantly changed. In contrast, a smaller parameter shift suggests that the pretrained representations are well-generalized and that the pretrained information is preserved during fine-tuning.

### 3.4 Scalability in Molecular SSL

In the fields of NLP and CV, SSL performance gradually improves as the amount of pretraining data or the number of model parameters increases Floridi & Chiriatti (2020); Kaplan et al. (2020); Zhai et al. (2022). Larger datasets offer models a wider variety of patterns, enabling them to learn more generalizable representations. As a result, scalability has become a key aspect of SSL. However, most of the prior papers considered in our study have not explored scalability. In this paper, we analyze how the size of the pretraining dataset influences the scalability of molecular SSL.

Specifically, we conduct experiments by changing only the size of the pretraining dataset, with the original model architectures kept as proposed in each paper. Our experiments use the ZINC15 dataset Sterling & Irwin (2015), which officially provides subsets containing 0.25 M and 2 M. Additionally, we create 0.02 M, 0.5 M, 1 M, and 1.5 M subsets by randomly sampling from the original 2 M dataset.

## 4 Experiments Setting

### 4.1 Datasets

**Pretraining Dataset.** We use 0.25 million unlabeled molecules from ZINC15 Sterling & Irwin (2015). Since pretraining does not aim to predict molecular properties, the data are randomly split into training and validation sets with a 9:1 ratio. The model is trained on the training set, and the checkpoint with the lowest validation loss is selected as the final pretrained model.

**Downstream Datasets** We use six molecular properties datasets from MoleculeNet Wu et al. (2018). BACE predicts whether a compound inhibits an enzyme. BBBP evaluates the ability of compounds to penetrate the blood-brain barrier. ClinTox is a binary classification task that distinguishes between FDA-approved drugs and compounds that failed clinical trials due to toxicity. Tox21 aims to predict the toxic effects of chemical compounds across multiple biological pathways. ToxCast provides detailed toxicity profiles across diverse biological and cellular pathways. SIDER includes information on drug side effects, covering 27 human organs. These datasets cover a variety of molecular and biological prediction tasks. Detailed information is provided in Table 3 in the Appendix.

### 4.2 Data Split

There are two common strategies for data splitting in molecular machine learning: random split and scaffold split. In domains such as computer vision and NLP, random splits are often used to evaluate out-of-distribution generalization. However, random splits are limited in the molecular domain because structurally or chemically similar molecules tend to exhibit similar properties Hendrickson (1991). Consequently, the model may have already seen test data patterns during training, leading to less reliable evaluation results. To address this issue, scaffold splitting is adopted Bemis & Murcko (1996). This method clusters molecules based on their unique core structures (scaffolds) and splits

Table 2: Performance on six downstream datasets and average with 3 repetitions under scaffold splitting, reported in terms of ROC-AUC (↑) as mean ± std in %. (A) Fine-tuning: starts from pretrained encoder weights, both the encoder and the prediction head are updated. (B) Linear probing: starts from pretrained encoder weights, the encoder is frozen, and only the prediction head is updated.

(A) Fine-tuning

| Method | BACE | BBBP | ClinTox | Tox21 | ToxCast | SIDER | AVG |
|---|---|---|---|---|---|---|---|
| GROVER | **85.93±1.18** | 92.73±3.60 | 84.90±6.71 | **84.91±2.05** | 62.41±0.69 | 70.33±1.27 | 80.20 |
| AttributeMask | 77.12±5.09 | 68.46±1.37 | 72.27±4.43 | 76.84±0.39 | 62.75±0.81 | 64.04±0.17 | 70.25 |
| ContextPred | 76.53±3.19 | 68.62±1.66 | 65.63±3.49 | 74.70±1.04 | 62.76±0.58 | 64.08±1.47 | 68.72 |
| EdgePred | 72.29±2.96 | 63.85±1.01 | 51.87±3.16 | 72.40±0.62 | 54.64±2.50 | 59.96±0.68 | 62.50 |
| GraphLoG | 83.51±0.76 | 63.13±1.34 | 63.78±4.76 | 73.26±0.39 | 60.39±0.69 | 62.64±0.84 | 67.79 |
| GraphCL | 78.83±1.31 | 63.84±0.51 | 58.59±4.79 | 73.17±0.79 | 60.13±0.16 | 63.00±1.51 | 66.26 |
| KANO | 84.73±2.18 | **94.61±1.14** | **88.08±4.32** | 83.52±2.52 | 59.36±1.33 | **72.41±2.19** | **80.45** |
| ChemBERTa | 77.24±1.20 | 78.12±1.04 | 85.73±6.45 | 70.75±1.92 | **69.73±1.47** | 52.23±2.78 | 72.30 |

(B) Linear probing

| Method | BACE | BBBP | ClinTox | Tox21 | ToxCast | SIDER | AVG |
|---|---|---|---|---|---|---|---|
| GROVER | **82.97±4.40** | 91.91±2.77 | **76.68±5.08** | **81.62±2.43** | 61.96±0.87 | 66.99±2.01 | **77.02** |
| AttributeMask | 61.76±0.69 | 60.09±0.56 | 65.27±1.82 | 69.55±0.23 | 54.56±0.67 | 57.65±1.29 | 61.48 |
| ContextPred | 60.07±1.58 | 63.43±0.16 | 23.49±0.55 | 68.29±0.44 | 60.77±0.82 | 58.21±0.69 | 55.71 |
| EdgePred | 63.36±7.09 | 56.57±1.03 | 49.91±0.49 | 51.60±2.07 | 51.51±0.46 | 49.96±0.40 | 53.82 |
| GraphLoG | 72.28±1.64 | 61.34±1.07 | 62.18±5.31 | 68.73±0.41 | 59.78±0.18 | 56.17±0.88 | 63.41 |
| GraphCL | 70.05±3.79 | 62.43±0.40 | 56.36±2.17 | 66.40±0.63 | 58.92±0.61 | 58.84±0.71 | 62.17 |
| KANO | 78.54±4.95 | **91.92±3.99** | 61.40±16.11 | 81.15±3.28 | 59.57±0.96 | **68.46±1.22** | 73.51 |
| ChemBERTa | 69.02±0.37 | 76.03±0.54 | 32.99±4.28 | 70.33±0.63 | **65.79±1.04** | 50.40±0.44 | 60.76 |

clusters into training, validation, and test sets. In our experiments, we use downstream datasets divided using scaffold splitting with an 8:1:1 ratio for the training, validation, and test sets.

## 4.3 IMPLEMENTATION DETAILS

The other hyperparameters for pretraining are set as follows: a batch size of 256, 100 epochs, and 300 hidden dimensions. For the downstream step, we use a batch size of 32, 50 epochs, and employ a 2-layer prediction head. We try to keep the original encoder structures and pretraining tasks unchanged. All the experiments are run on a single NVIDIA RTX 3090 GPU.

## 5 RESULT

### 5.1 ANALYZING GENERALIZATION VIA FINE-TUNING AND LINEAR PROBING

We design a unified experimental setup to focus on pretraining. We conduct fine-tuning, and the results are presented in Table 2 (A). KANO achieves the highest average performance (80.45), followed closely by GROVER (80.20), indicating that their performance is comparable.

Pretrained representations are modified during fine-tuning to fit downstream tasks, which can make it difficult to accurately assess the quality of the original pretrained representations. To address this, we adapt linear probing, which preserves the pretrained representations, and shows the results in Table 2 (B). GROVER achieves the highest performance in linear probing (77.02), suggesting that its pretrained representations are reasonably general, by showing high performance across diverse tasks without encoder updates. KANO achieves the highest performance in fine-tuning, which leads to the common expectation that its pretrained representations are the most generalizable. However, KANO ranks second in linear probing, implying that its pretrained representations may be slightly less generalizable than GROVER.

To further assess the generality of the pretrained representations, we compare the performance of fine-tuning and linear probing. The results are shown in Figure 6 in the Appendix. The performance gap between fine-tuning and linear probing indicates how effectively the pretrained encoder can be

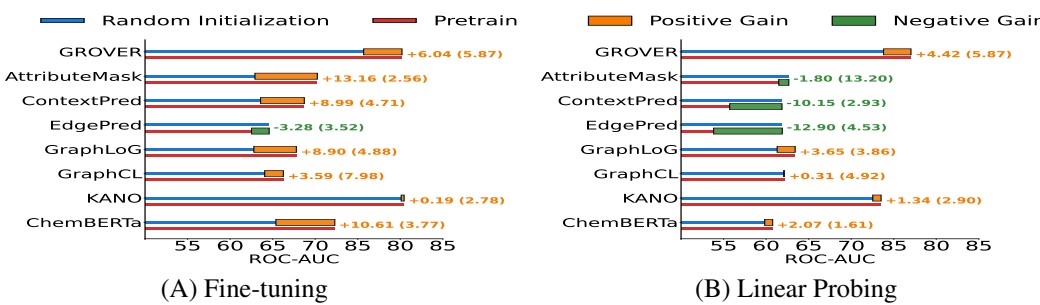

(A) Fine-tuning  (B) Linear Probing

Figure 2: Comparison of Pretrain Gain under (A) fine-tuning and (B) linear probing. Each line bar represents the average ROC-AUC across six downstream datasets with 3 repetitions, the red and blue indicate pretrain and randomly initialized, respectively. Pretrain Gain is represented using rectangular bars, with positive gain in orange and negative in green, with values of mean and standard deviation.

utilized in downstream tasks without modification. GROVER, GraphLoG, and GraphCL exhibit a low performance gap of less than 5, suggesting that their pretrained representations are well-generalized. In contrast, ContextPred and ChemBERTa exhibit a performance gap of over 10, indicating a substantial drop in performance when the pretrained representations are used without modification. This may imply that their representations are less generalizable under our evaluation setup, where a smaller performance gap indicates more generalizable and robust pretrained representations. Therefore, designing pretraining tasks that reduce the gap between fine-tuning and linear probing is desirable, as it may lead to more robust and generalizable molecular representations.

## 5.2 ASSESSING THE CONTRIBUTION OF PRETRAINED REPRESENTATIONS THROUGH PRETRAIN GAIN

To quantify the performance improvement achieved through pretraining, we use Pretrain Gain. Figure 2 is computed based on the results shown in Table 2 and Table 4 in the Appendix. A positive Pretrain Gain suggests that pretraining provides a benefit, resulting in better performance than a randomly initialized model. The Pretrain Gain under fine-tuning is shown in Figure 2 (A). Most models show a positive Pretrain Gain, which is consistent with prior work demonstrating the benefits of pretraining. Interestingly, KANO — despite achieving the highest fine-tuning performance — shows a negligible Pretrain Gain (0.34), suggesting that the high performance may not be due to pretraining. This result highlights that the fine-tuning result alone is insufficient to assess the effect of pretraining, emphasizing the importance of Pretrain Gain as an evaluation metric

As shown in Figure 2 (B), which presents the Pretrain Gain under linear probing, the results substantially differ from the trends observed in fine-tuning. Most models show a positive Pretrain Gain in fine-tuning; however, the Pretrain Gain in linear probing is smaller than the Pretrain Gain observed in fine-tuning. Specifically, except for ChemBERTa, no model exceeds a Pretrain Gain of 5%. Surprisingly, in some cases, randomly initialized models outperform the pretrained model. These observations suggest that the pretrained representations may not have captured sufficiently transferable features for linear probing. These results show that even if a model achieves high performance in fine-tuning, it does not always imply high-quality representations.

## 5.3 QUANTIFYING FORGETTING VIA PARAMETER SHIFT

We measure parameter shift to quantify forgetting during fine-tuning, as summarized in Table 5 in the Appendix and illustrated in Figure 3. GROVER and ChemBERTa, both Transformer-based models, exhibit relatively small parameter shifts, suggesting that their pretrained representations are sufficiently general and well preserved during fine-tuning. In contrast, GNN-based models tend to exhibit substantial parameter shifts, particularly in tasks such as Tox21 and ToxCast. As shown in Table 3 in the Appendix, as these datasets are larger and more diverse, this may increase the need for generalized representations. If the pretrained representations fail to capture such molecular diversity, the model may require more substantial parameter updates during fine-tuning.

Traditional GNN-based models design pretext tasks that focus on learning the structural information of molecular graphs. However, downstream tasks often require a deeper understanding of chemical properties, leading to a discrepancy between pretraining and the downstream task. KANO addresses this issue through a prompt-based mechanism that incorporates functional prompts extracted from a knowledge graph, enabling the model to learn both structural and chemical knowledge during pretraining. This design is intended to reduce the discrepancy between pretraining and downstream tasks. Interestingly, although GNN-based models typically show significant parameter shifts during fine-tuning, KANO exhibits relatively small shifts, which may suggest that forgetting of pretrained knowledge is mitigated. This observation implies that adopting Transformer-based architectures or leveraging knowledge graphs can help reduce parameter shifts and preserve pretrained knowledge more effectively.

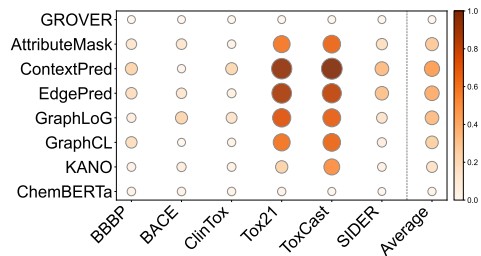

Figure 3: Quantification of encoder parameter shifts due to fine-tuning. Circle size and color represent the mean and variance of parameter shifts, respectively. Darker colors and larger circles represent greater parameter changes, while lighter colors and smaller circles indicate smaller changes.

We compare the performance gap—used as a measure of generality—with the ranking of parameter shift. As shown in Figure 7 of the appendix, the two metrics exhibit an linear relationship: larger parameter shifts correspond to larger performance gaps. A larger performance gap indicates weaker generalization, suggesting that models with larger parameter shifts produce less generalizable representations. Thus, parameter shift provides a useful indirectly metric for evaluating representation generality.

## 5.4 SCALABILITY OF MOLECULAR SSL

Figure 4 visualizes the average performance reported in Table6–10 in the Appendix, which presents results under varying pretraining dataset sizes to analyze scalability. Most models exhibit a flat performance trend regardless of the amount of pretraining data. This pattern is observed in both fine-tuning and linear probing results, suggesting that these models have limited scalability under our experimental setting.

We consider one main factor to understand this limitation. Unlike the NLP and CV domains Hoffmann et al. (2022), molecular data is characterized by subtle structural diversity and domain-specific constraints. Existing molecular pretraining methods, such as masking and contrastive learning, aim to capture chemically meaningful information through structural perturbations. However, structure-based approaches may be insufficient to capture certain chemical properties of molecules, especially those not directly linked to graph structure. Therefore, overcoming this limitation require pre-

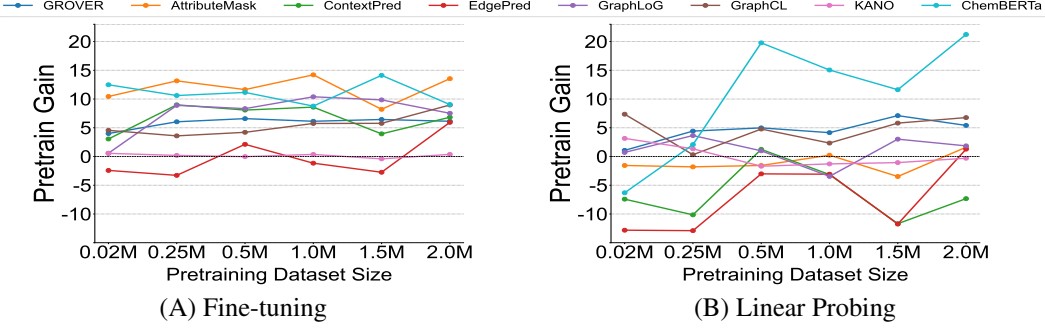

Figure 4: Pretrain Gain (%) across varying pretraining dataset sizes for eight molecular SSL models under (A) fine-tuning and (B) linear probing. Pretrain Gain is averaged over six downstream tasks, each repeated three times, for each pretraining dataset size.

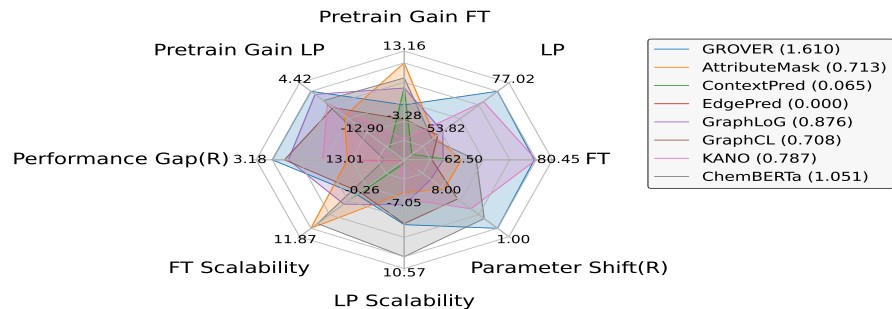

Figure 5: The graph illustrates model performance across eight evaluation settings using a polygon representation. Fine-tuning and linear probing are denoted as FT and LP, respectively. For metrics marked with (R), lower values indicate better performance, so they are computed in reverse order. Scalability is caculated by averaging results across datasets, while Parameter Shift use ranking. A larger polygonal area indicates stronger performance. In the legend, the value next to each model denotes its polygon area.

training strategies that reduce the discrepancy between pretraining and downstream tasks, enabling performance to scale with larger datasets.

## 5.5 Integrated Evaluation Results

As shown in Figure 5, we present a comprehensive evaluation integrating eight methods for quantitative comparison. A key observation is that no model achieves balanced performance across all eight metrics. Nevertheless, GROVER emerges as the strongest overall model, excelling in most metrics except for Pretrain Gain FT and FT scalability. KANO achieves the highest performance under the widely adopted fine-tuning but performs poorly in both Pretrain Gain and scalability, leading to an overall ranking of fourth. This demonstrates that strong fine-tuning performance does not guarantee overall superiority in pretraining approaches.

Taken together, our results indicate that Transformer-based architectures are particularly effective, with GROVER and ChemBERTa achieving the highest overall performance. For GNN-based models, contrastive learning generally proves to be a strong pretraining strategy, with GraphLoG and KANO achieving the best performance among GNNs. However, GraphCL performs worse than AttributeMask, suggesting that basic contrastive learning alone is insufficient and that more advanced strategies are required.

To further validate our findings, we provide additional results in the appendix. The regression results in Table 11–16 and Figure 8, 9 show that scalability remains flat, while the experiments with a hidden dimension of 1200 (Table 17, Figure 10) reveal that linear probing yields more negative gains. These results are consistent with our main findings, thereby reinforcing the robustness of our conclusions.

## 6 Conclusion

In this paper, we present a multi-perspective evaluation framework for molecular SSL beyond fine-tuning, incorporating linear probing, Pretrain Gain, parameter shift analysis, scalability. Our results reveal that high fine-tuning performance does not necessarily imply generalizable pretrained representations, highlighting the limitations of relying solely on fine-tuning for evaluation. Through parameter shift analysis, we show that GNN-based models encounter substantial parameter shifts during fine-tuning, raising concerns about the stability and generality of their representations. We also find that many models exhibit limited scalability, with flat trends from larger pretraining datasets, unlike trends observed in NLP and CV. In the comprehensive evaluation, no model achieves consistently high performance across all metrics, underscoring the limited generalization of molecular SSL representations. This suggests that advancing molecular graph SSL requires moving beyond a focus solely on fine-tuning accuracy and should adopt comprehensive evaluation frameworks such as the one proposed in this paper.

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

## 7 APPENDIX

Table 3: Details of the dataset used in the experiments. # Tasks and # Compounds are the number of tasks to perform and molecules, respectively. # Atoms and # Bonds are the averages of the number of nodes and edges in all molecules, respectively.

| DATASET | # TASKS | # GRAPHS | # ATOMS | # BONDS |
|---------|---------|----------|---------|---------|
| BACE | 1 | 1,513 | 34.1 | 36.9 |
| BBBP | 1 | 2,03 | 24.1 | 26.0 |
| CLINTOX | 2 | 1,478 | 26.3 | 28.1 |
| TOX21 | 12 | 7,831 | 18.6 | 19.3 |
| SIDER | 27 | 1,478 | 34.3 | 36.1 |
| TOXCAST | 617 | 8,575 | 18.8 | 19.3 |

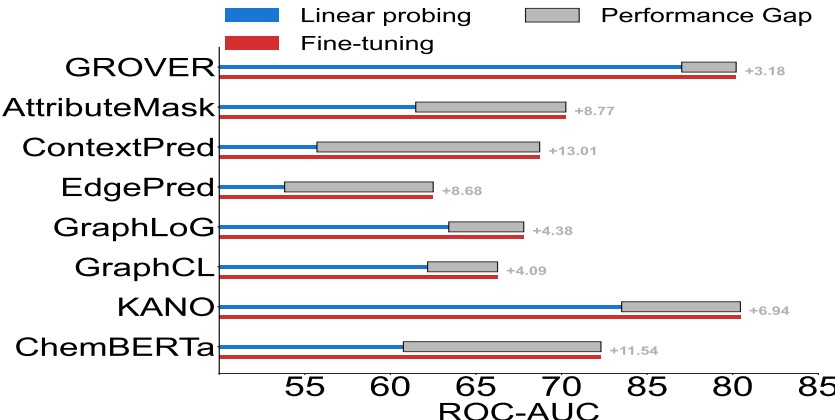

Figure 6: Figure illustrates the performance gap between fine-tuning and linear probing. A smaller gap indicates that linear probing achieves high performance, suggesting that the pretrained representations are highly generalizable.

Table 4: Prediction performance on six downstream tasks and the overall average (across 3 repeats) using scaffold splitting, reported in terms of ROC-AUC ($\uparrow$) as mean and std in %. (A) Random Initialization (Fine-tuning): Starts from randomly initialized encoder weights; both the encoder and the prediction head are trained. (B) Random Initialization (Linear Probing): Starts from randomly initialized encoder weights; the encoder is frozen, and only the prediction head is trained.

| (A) Random Initialization (Fine-tuning) | | | | | | |
|---|---|---|---|---|---|---|
| Method | BACE | BBBP | ClinTox | Tox21 | ToxCast | SIDER | AVG |
| GROVER | $79.14_{\pm 5.19}$ | $91.51_{\pm 2.85}$ | $74.95_{\pm 4.89}$ | $81.60_{\pm 2.07}$ | $65.56_{\pm 1.59}$ | $61.36_{\pm 2.82}$ | 75.69 |
| AttributeMask | $70.52_{\pm 2.50}$ | $66.78_{\pm 0.94}$ | $53.12_{\pm 3.23}$ | $73.11_{\pm 0.98}$ | $61.75_{\pm 0.73}$ | $59.11_{\pm 0.39}$ | 64.07 |
| ContextPred | $66.07_{\pm 3.75}$ | $68.34_{\pm 1.05}$ | $49.10_{\pm 6.12}$ | $73.06_{\pm 0.95}$ | $61.35_{\pm 1.53}$ | $59.57_{\pm 3.49}$ | 62.92 |
| EdgePred | $72.69_{\pm 6.11}$ | $66.26_{\pm 2.22}$ | $51.47_{\pm 6.95}$ | $73.06_{\pm 0.36}$ | $60.93_{\pm 1.02}$ | $56.98_{\pm 1.82}$ | 63.57 |
| GraphLoG | $74.74_{\pm 2.36}$ | $67.99_{\pm 1.35}$ | $54.26_{\pm 1.57}$ | $72.58_{\pm 0.81}$ | $61.75_{\pm 1.13}$ | $56.20_{\pm 2.96}$ | 64.59 |
| GraphCL | $71.09_{\pm 3.84}$ | $65.20_{\pm 3.84}$ | $48.74_{\pm 0.82}$ | $73.76_{\pm 1.05}$ | $61.92_{\pm 0.53}$ | $56.05_{\pm 1.47}$ | 62.79 |
| KANO | $84.35_{\pm 0.56}$ | $93.50_{\pm 2.82}$ | $85.36_{\pm 5.17}$ | $83.44_{\pm 2.29}$ | $71.66_{\pm 1.17}$ | $62.36_{\pm 2.04}$ | **80.11** |
| ChemBERTa | $71.17_{\pm 3.11}$ | $72.02_{\pm 4.02}$ | $64.16_{\pm 8.13}$ | $66.48_{\pm 2.74}$ | $68.89_{\pm 1.89}$ | $49.56_{\pm 2.99}$ | 65.38 |
| (B) Random Initialization (Linear probing) | | | | | | |
| Method | BACE | BBBP | ClinTox | Tox21 | ToxCast | SIDER | AVG |
| Grover | $82.97_{\pm 4.40}$ | $91.91_{\pm 2.77}$ | $76.68_{\pm 5.08}$ | $81.62_{\pm 2.43}$ | $66.99_{\pm 0.87}$ | $61.96_{\pm 2.01}$ | **77.02** |
| AttributeMask | $61.76_{\pm 0.69}$ | $60.09_{\pm 0.56}$ | $65.27_{\pm 1.82}$ | $69.55_{\pm 0.23}$ | $57.65_{\pm 0.67}$ | $54.56_{\pm 1.29}$ | 61.48 |
| ContextPred | $60.07_{\pm 1.58}$ | $63.43_{\pm 0.16}$ | $23.49_{\pm 0.55}$ | $68.29_{\pm 0.44}$ | $58.21_{\pm 0.82}$ | $60.77_{\pm 0.69}$ | 55.71 |
| EdgePred | $63.36_{\pm 7.09}$ | $56.57_{\pm 1.03}$ | $49.91_{\pm 0.49}$ | $51.60_{\pm 2.07}$ | $51.51_{\pm 0.46}$ | $49.96_{\pm 0.40}$ | 53.82 |
| GraphLog | $72.28_{\pm 1.64}$ | $61.34_{\pm 1.07}$ | $62.18_{\pm 5.31}$ | $68.73_{\pm 0.41}$ | $56.17_{\pm 0.18}$ | $59.78_{\pm 0.88}$ | 63.41 |
| GraphCL | $70.05_{\pm 3.79}$ | $62.43_{\pm 0.40}$ | $56.36_{\pm 2.17}$ | $66.40_{\pm 0.63}$ | $58.84_{\pm 0.61}$ | $58.92_{\pm 0.71}$ | 62.17 |
| KANO | $78.54_{\pm 4.95}$ | $91.92_{\pm 3.99}$ | $61.40_{\pm 16.11}$ | $81.15_{\pm 3.28}$ | $68.46_{\pm 0.96}$ | $59.57_{\pm 1.22}$ | 73.51 |
| ChemBERTa | $69.02_{\pm 0.37}$ | $76.03_{\pm 0.54}$ | $32.99_{\pm 4.28}$ | $70.33_{\pm 0.63}$ | $65.79_{\pm 1.05}$ | $50.40_{\pm 0.44}$ | 60.76 |

The table shows the numerical values of the parameter shifts visualized in Figure 3.

Table 5: This table shows the mean and standard deviation of L2-based parameter shifts for each dataset.

| Method | BACE | BBBP | ClinTox | Tox21 | ToxCast | SIDER | AVG |
|---|---|---|---|---|---|---|---|
| GROVER | 150.56 ±6.79 | 114.52 ±5.01 | 14.47 ±0.72 | 146.88 ±7.99 | 83.40±4.08 | 202.52 ±10.55 | 118.73 |
| AttributeMask | 7342.73 ±324.47 | 7512.03 ±342.73 | 1621.39 ±81.04 | 33802.75±1467.07 | 37624.28±1623.60 | 9871.82 ±460.89 | 16295.83 |
| ContextPred | 13259.14±585.36 | 959.35 ±42.16 | 12196.34±567.81 | 53328.02±2256.96 | 56409.84±2435.61 | 19260.63±841.49 | **25902.22** |
| EdgePred | 10292.06±487.97 | 7128.03 ±337.83 | 2880.88 ±133.83 | 49563.34±2209.74 | 46475.73±2074.52 | 18025.51±856.93 | 22394.26 |
| GraphLoG | 3575.96 ±189.63 | 13064.01±707.52 | 8800.22 ±482.51 | 41243.57±2083.18 | 39080.37±1757.49 | 8217.41 ±358.15 | 18996.92 |
| GraphCL | 10232.89±531.38 | 351.13 ±15.59 | 1395.80 ±63.51 | 34420.74±1581.78 | 37580.58±1720.92 | 3829.59 ±174.62 | 14635.12 |
| KANO | 2406.61 ±109.54 | 3817.20 ±190.18 | 2454.36 ±110.82 | 13678.55±701.49 | 28898.60 ±1502.89 | 2227.76 ±103.35 | 8913.85 |
| ChemBERTa | 1777.71 ±24.31 | 1649.05±22.61 | 1463.93 ±20.10 | 1393.99 ±19.26 | 553.03 ±7.87 | 31.30 ±0.61 | 1144.83 |

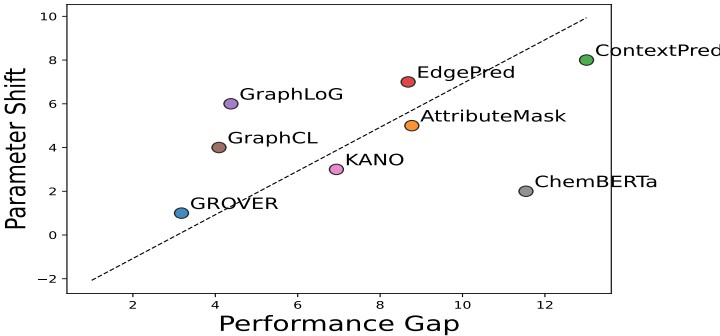

Figure 7: The relationship between parameter shift—calculated based on ranking—and the performance gap, which reflects the generality of pretrained representations.

These tables show the performance on each dataset size from the scalability experiment.

Table 6: Fine-tuning and linear probing results of models pretrained on 0.02M dataset.

| Fine-tuning | | | | | | | |
|---|---|---|---|---|---|---|---|
| Method | BACE | BBBP | ClinTox | Tox21 | ToxCast | SIDER | AVG |
| Grover | 85.07±2.09 | 93.10±3.99 | 79.04±9.67 | 83.81±1.01 | 69.38±0.53 | 61.83±2.16 | 78.70 |
| AttributeMask | 82.18±3.36 | 69.86±1.89 | 63.87±4.96 | 74.89±0.99 | 64.36±0.88 | 58.73±1.75 | 68.98 |
| ContextPred | 75.99±6.20 | 68.12±0.39 | 50.13±4.58 | 74.71±0.52 | 62.22±0.37 | 62.21±0.07 | 65.57 |
| EdgePred | 70.26±3.01 | 67.01±3.37 | 49.43±4.00 | 73.46±0.24 | 60.83±0.55 | 57.38±1.44 | 63.06 |
| GraphLog | 69.31±10.44 | 63.08±4.04 | 51.87±4.24 | 72.88±0.14 | 62.27±1.15 | 57.96±1.06 | 62.89 |
| GraphCL | 78.14±4.02 | 68.11±1.64 | 58.17±3.26 | 74.32±0.89 | 64.06±0.80 | 58.85±2.71 | 66.94 |
| KANO | 83.52±1.92 | 93.86±3.64 | 87.59±4.21 | 83.66±2.52 | 72.39±0.97 | 62.34±1.44 | **80.56** |
| ChemBERTa | 85.48±0.59 | 69.85±2.45 | 99.27±0.11 | 65.20±1.49 | 61.11±2.13 | 57.63±1.94 | 73.09 |
| Linear Probing | | | | | | | |
| Method | BACE | BBBP | ClinTox | Tox21 | ToxCast | SIDER | AVG |
| Grover | 82.26±2.34 | 92.62±4.32 | 67.76±5.92 | 80.98±1.58 | 67.71±0.61 | 61.52±0.51 | **75.47** |
| AttributeMask | 77.71±1.81 | 60.37±0.77 | 44.99±6.56 | 68.67±0.59 | 60.30±0.59 | 58.14±0.83 | 61.70 |
| ContextPred | 63.11±0.33 | 62.81±0.87 | 40.95±4.43 | 62.54±0.63 | 58.61±1.20 | 55.54±0.73 | 57.26 |
| EdgePred | 62.35±3.87 | 53.97±1.18 | 51.43±2.79 | 51.98±0.33 | 51.84±0.79 | 51.21±0.87 | 53.80 |
| GraphLog | 69.26±1.69 | 57.18±1.59 | 54.22±2.03 | 69.40±0.48 | 58.78±0.36 | 61.07±1.99 | 61.65 |
| GraphCL | 68.25±1.75 | 66.19±0.71 | 73.74±4.52 | 71.97±0.25 | 58.50±1.06 | 59.98±0.42 | 66.44 |
| KANO | 82.61±5.01 | 88.67±1.94 | 69.45±10.04 | 80.20±3.58 | 67.62±1.40 | 58.74±2.74 | 74.55 |
| ChemBERTa | 57.13±3.32 | 59.92±4.70 | 61.24±27.96 | 51.11±2.83 | 49.65±0.19 | 49.99±0.47 | 54.84 |

Table 7: Fine-tuning and linear probing results of models pretrained on 0.5M dataset.
head are trained.

**Fine-tuning**

| Method | BACE | BBBP | ClinTox | Tox21 | ToxCast | SIDER | AVG |
|---|---|---|---|---|---|---|---|
| Grover | $86.99_{\pm1.18}$ | $92.94_{\pm4.51}$ | $85.14_{\pm5.10}$ | $85.31_{\pm2.46}$ | $70.61_{\pm0.80}$ | $62.71_{\pm1.87}$ | **80.62** |
| AttributeMask | $79.01_{\pm1.02}$ | $67.54_{\pm0.92}$ | $68.49_{\pm6.70}$ | $74.93_{\pm1.46}$ | $64.20_{\pm0.15}$ | $62.37_{\pm0.67}$ | 70.74 |
| ContextPred | $81.30_{\pm1.12}$ | $69.28_{\pm1.42}$ | $60.19_{\pm4.61}$ | $75.07_{\pm0.94}$ | $64.19_{\pm0.30}$ | $60.98_{\pm1.68}$ | 68.50 |
| EdgePred | $76.38_{\pm5.54}$ | $65.51_{\pm2.98}$ | $54.37_{\pm4.98}$ | $73.63_{\pm1.02}$ | $63.25_{\pm0.56}$ | $61.86_{\pm0.79}$ | 65.83 |
| GraphLog | $81.19_{\pm1.51}$ | $67.53_{\pm2.52}$ | $62.73_{\pm3.86}$ | $74.00_{\pm0.46}$ | $61.58_{\pm1.03}$ | $58.13_{\pm0.12}$ | 67.29 |
| GraphCL | $70.02_{\pm2.17}$ | $68.41_{\pm1.05}$ | $63.92_{\pm4.09}$ | $73.23_{\pm0.81}$ | $63.45_{\pm0.23}$ | $59.26_{\pm1.07}$ | 66.38 |
| KANO | $83.48_{\pm2.08}$ | $93.93_{\pm2.30}$ | $88.19_{\pm8.10}$ | $83.87_{\pm2.00}$ | $72.12_{\pm1.19}$ | $59.87_{\pm0.18}$ | 80.24 |
| ChemBERTa | $76.76_{\pm4.07}$ | $81.97_{\pm2.46}$ | $88.70_{\pm0.91}$ | $66.45_{\pm2.49}$ | $72.12_{\pm0.82}$ | $50.71_{\pm0.79}$ | 72.79 |

**Linear Probing**

| Method | BACE | BBBP | ClinTox | Tox21 | ToxCast | SIDER | AVG |
|---|---|---|---|---|---|---|---|
| Grover | $82.60_{\pm4.09}$ | $92.69_{\pm2.74}$ | $77.85_{\pm7.65}$ | $81.59_{\pm2.28}$ | $67.87_{\pm0.67}$ | $62.00_{\pm1.94}$ | **77.43** |
| AttributeMask | $67.66_{\pm12.23}$ | $59.30_{\pm0.49}$ | $60.20_{\pm2.99}$ | $68.13_{\pm0.44}$ | $59.57_{\pm0.24}$ | $58.01_{\pm1.23}$ | 61.65 |
| ContextPred | $73.82_{\pm3.49}$ | $63.98_{\pm1.41}$ | $50.68_{\pm0.77}$ | $69.17_{\pm0.73}$ | $60.18_{\pm0.91}$ | $58.39_{\pm0.51}$ | 62.70 |
| EdgePred | $61.03_{\pm1.30}$ | $58.20_{\pm0.38}$ | $55.75_{\pm2.22}$ | $67.69_{\pm0.42}$ | $57.82_{\pm0.09}$ | $58.51_{\pm0.43}$ | 59.83 |
| GraphLog | $70.00_{\pm2.30}$ | $60.41_{\pm0.98}$ | $61.28_{\pm4.17}$ | $66.56_{\pm0.05}$ | $55.97_{\pm0.30}$ | $56.38_{\pm0.74}$ | 61.77 |
| GraphCL | $74.80_{\pm1.50}$ | $62.32_{\pm1.47}$ | $65.78_{\pm3.70}$ | $67.51_{\pm1.13}$ | $59.47_{\pm0.52}$ | $59.58_{\pm0.89}$ | 64.91 |
| KANO | $77.19_{\pm7.76}$ | $91.87_{\pm2.98}$ | $53.39_{\pm10.40}$ | $80.04_{\pm2.99}$ | $68.65_{\pm1.15}$ | $58.29_{\pm0.99}$ | 71.57 |
| ChemBERTa | $74.08_{\pm0.39}$ | $76.66_{\pm0.91}$ | $89.61_{\pm1.89}$ | $63.42_{\pm0.20}$ | $65.92_{\pm0.07}$ | $52.12_{\pm1.04}$ | 70.30 |

Table 8: Fine-tuning and linear probing results of models pretrained on 1 M dataset.

**Fine-tuning**

| Method | BACE | BBBP | ClinTox | Tox21 | ToxCast | SIDER | AVG |
|---|---|---|---|---|---|---|---|
| Grover | $85.68_{\pm1.70}$ | $93.01_{\pm3.56}$ | $85.77_{\pm4.18}$ | $84.87_{\pm2.10}$ | $70.41_{\pm0.28}$ | $61.98_{\pm0.87}$ | 80.29 |
| AttributeMask | $78.59_{\pm0.85}$ | $69.41_{\pm3.44}$ | $76.48_{\pm4.71}$ | $75.98_{\pm0.63}$ | $63.99_{\pm0.64}$ | $59.96_{\pm1.56}$ | 70.74 |
| ContextPred | $74.51_{\pm9.06}$ | $69.84_{\pm5.46}$ | $64.76_{\pm0.51}$ | $74.88_{\pm0.53}$ | $64.05_{\pm0.52}$ | $62.78_{\pm0.18}$ | 68.47 |
| EdgePred | $67.08_{\pm4.13}$ | $69.24_{\pm0.47}$ | $55.75_{\pm2.80}$ | $73.63_{\pm0.52}$ | $60.62_{\pm0.46}$ | $55.70_{\pm1.39}$ | 63.67 |
| GraphLog | $82.80_{\pm1.68}$ | $66.52_{\pm0.69}$ | $68.01_{\pm3.71}$ | $73.97_{\pm1.10}$ | $61.89_{\pm0.31}$ | $58.33_{\pm1.20}$ | 68.59 |
| GraphCL | $81.74_{\pm2.08}$ | $67.36_{\pm0.19}$ | $58.49_{\pm4.59}$ | $73.79_{\pm0.58}$ | $63.80_{\pm0.49}$ | $61.08_{\pm0.99}$ | 67.71 |
| KANO | $84.94_{\pm0.56}$ | $94.29_{\pm1.80}$ | $87.53_{\pm7.46}$ | $83.32_{\pm2.24}$ | $72.28_{\pm1.47}$ | $60.72_{\pm0.88}$ | **80.51** |
| ChemBERTa | $76.33_{\pm1.42}$ | $79.54_{\pm1.41}$ | $75.23_{\pm5.15}$ | $70.04_{\pm0.84}$ | $69.66_{\pm2.92}$ | $55.11_{\pm0.88}$ | 70.99 |

**Linear Probing**

| Method | BACE | BBBP | ClinTox | Tox21 | ToxCast | SIDER | AVG |
|---|---|---|---|---|---|---|---|
| Grover | $82.16_{\pm4.16}$ | $92.77_{\pm2.74}$ | $75.78_{\pm7.16}$ | $81.24_{\pm2.58}$ | $67.58_{\pm0.71}$ | $61.48_{\pm2.16}$ | **76.84** |
| AttributeMask | $69.64_{\pm0.47}$ | $60.58_{\pm0.30}$ | $68.63_{\pm3.06}$ | $66.38_{\pm0.10}$ | $60.38_{\pm0.45}$ | $51.37_{\pm0.91}$ | 62.83 |
| ContextPred | $69.43_{\pm2.91}$ | $59.86_{\pm2.39}$ | $41.91_{\pm3.06}$ | $68.70_{\pm1.13}$ | $59.09_{\pm0.51}$ | $60.80_{\pm1.08}$ | 59.96 |
| EdgePred | $61.03_{\pm1.30}$ | $58.20_{\pm0.38}$ | $55.66_{\pm2.25}$ | $67.64_{\pm0.43}$ | $57.75_{\pm0.18}$ | $58.48_{\pm0.41}$ | 59.79 |
| GraphLog | $66.87_{\pm2.39}$ | $53.08_{\pm0.42}$ | $57.64_{\pm3.98}$ | $65.78_{\pm0.13}$ | $55.83_{\pm0.34}$ | $54.87_{\pm0.50}$ | 59.01 |
| GraphCL | $72.32_{\pm0.82}$ | $64.89_{\pm1.02}$ | $56.20_{\pm3.59}$ | $67.39_{\pm0.64}$ | $60.47_{\pm0.40}$ | $59.47_{\pm1.23}$ | 63.46 |
| KANO | $72.42_{\pm12.67}$ | $92.11_{\pm3.09}$ | $59.60_{\pm12.22}$ | $80.39_{\pm3.34}$ | $68.38_{\pm1.01}$ | $57.12_{\pm2.33}$ | 71.67 |
| ChemBERTa | $64.30_{\pm9.50}$ | $76.54_{\pm1.23}$ | $80.59_{\pm3.77}$ | $65.28_{\pm0.59}$ | $64.99_{\pm0.13}$ | $52.47_{\pm0.59}$ | 67.36 |

Table 9: Fine-tuning and linear probing results of models pretrained on 1.5 M dataset.

Fine-tuning

| Method | BACE | BBBP | ClinTox | Tox21 | ToxCast | SIDER | AVG |
|---|---|---|---|---|---|---|---|
| Grover | 84.39±2.80 | 92.45±4.87 | 86.21±5.23 | 85.48±2.41 | 71.81±0.88 | 62.39±0.95 | **80.45** |
| AttributeMask | 81.06±2.21 | 67.96±3.82 | 59.34±2.24 | 75.02±0.61 | 63.41±0.10 | 59.71±0.80 | 67.75 |
| ContextPred | 75.48±3.42 | 67.81±2.08 | 56.73±9.11 | 73.85±0.28 | 61.51±0.95 | 59.94±1.76 | 65.89 |
| EdgePred | 68.73±6.20 | 66.79±2.01 | 50.31±4.67 | 72.07±0.51 | 62.15±0.54 | 56.60±1.50 | 62.77 |
| GraphLog | 82.98±0.57 | 65.22±2.42 | 62.63±1.76 | 74.44±0.05 | 63.21±0.62 | 62.11±1.75 | 68.43 |
| GraphCL | 78.76±1.23 | 67.58±2.15 | 60.25±1.93 | 75.41±0.66 | 63.44±1.04 | 60.55±1.37 | 67.66 |
| KANO | 83.62±1.31 | 94.53±1.74 | 84.02±4.57 | 83.04±1.76 | 72.94±1.06 | 60.83±1.50 | 79.83 |
| ChemBERTa | 71.05±0.83 | 87.42±1.84 | 98.55±0.19 | 66.93±0.99 | 62.12±0.95 | 58.98±0.20 | 74.18 |

Linear Probing

| Method | BACE | BBBP | ClinTox | Tox21 | ToxCast | SIDER | AVG |
|---|---|---|---|---|---|---|---|
| Grover | 83.80±2.24 | 93.05±4.15 | 83.19±7.27 | 82.26±3.05 | 68.32±0.81 | 63.36±1.78 | **79.00** |
| AttributeMask | 60.63±2.24 | 64.13±0.16 | 55.65±1.96 | 69.63±0.45 | 59.10±0.21 | 53.23±3.39 | 60.40 |
| ContextPred | 42.43±4.47 | 59.38±0.47 | 43.16±6.56 | 65.20±0.17 | 57.89±0.35 | 57.60±0.38 | 54.28 |
| EdgePred | 69.32±4.57 | 56.38±2.31 | 49.24±4.16 | 51.73±0.57 | 51.93±0.38 | 49.17±2.65 | 54.63 |
| GraphLog | 74.10±1.26 | 62.36±1.18 | 57.52±0.29 | 69.14±0.26 | 56.81±0.50 | 59.10±0.91 | 63.17 |
| GraphCL | 73.34±1.12 | 66.66±0.86 | 66.27±2.38 | 69.80±0.90 | 59.33±1.04 | 58.25±1.12 | 65.61 |
| KANO | 75.31±7.02 | 92.04±2.84 | 57.61±16.89 | 80.51±2.76 | 68.29±0.72 | 57.73±1.51 | 71.92 |
| chemberta | 78.02±4.55 | 80.15±2.00 | 39.27±6.63 | 73.93±0.85 | 73.90±1.80 | 54.24±1.55 | 66.59 |

Table 10: Fine-tuning and linear probing results of models pretrained on 2 M dataset.

Fine-tuning

| Method | BACE | BBBP | ClinTox | Tox21 | ToxCast | SIDER | AVG |
|---|---|---|---|---|---|---|---|
| Grover | 85.68±1.70 | 93.01± 3.56 | 85.77±4.18 | 84.87±2.10 | 70.41±0.28 | 61.98±0.87 | 80.29 |
| AttributeMask | 80.81±2.53 | 70.12± 1.12 | 71.58±7.19 | 75.75±0.65 | 63.47±0.82 | 61.61±1.01 | 70.56 |
| ContextPred | 76.78±10.81 | 67.88± 0.91 | 59.26±3.05 | 74.54±0.51 | 64.31±0.64 | 62.78±1.76 | 67.59 |
| EdgePred | 76.30±8.16 | 69.86± 10.91 | 61.75±2.69 | 75.69±0.18 | 64.27±0.45 | 61.05±0.21 | 68.15 |
| GraphLog | 79.01±11.22 | 67.66± 2.71 | 59.69±3.15 | 73.29±0.42 | 62.07±0.57 | 60.60±1.30 | 67.05 |
| GraphCL | 78.15±3.26 | 68.13± 0.27 | 72.93±3.27 | 74.61±0.07 | 63.71±0.12 | 58.17±1.48 | 69.28 |
| KANO | 84.02±1.38 | 93.71± 1.68 | 87.13±8.28 | 84.11±1.54 | 72.63±1.51 | 61.21±1.02 | **80.47** |
| ChemBERTa | 79.02±2.25 | 79.19± 3.19 | 66.21±21.11 | 73.75±1.03 | 72.12±1.51 | 56.71±1.09 | 71.17 |

Linear Probing

| Method | BACE | BBBP | ClinTox | Tox21 | ToxCast | SIDER | AVG |
|---|---|---|---|---|---|---|---|
| Grover | 82.80±3.99 | 92.11±2.52 | 81.29±4.44 | 81.40±2.60 | 67.35±0.68 | 61.74±3.13 | **77.78** |
| AttributeMask | 62.55±0.27 | 66.46±0.90 | 73.10±0.77 | 69.24±0.50 | 56.57±0.27 | 54.29±0.24 | 63.70 |
| ContextPred | 65.52±7.00 | 60.88±0.59 | 30.00±1.09 | 68.77±0.52 | 60.32±0.78 | 59.22±0.26 | 57.45 |
| EdgePred | 70.81±2.73 | 59.92±1.32 | 64.68±2.77 | 64.69±0.95 | 59.01±0.49 | 56.13±1.12 | 62.54 |
| GraphLog | 71.78±1.98 | 58.66±0.69 | 59.57±1.74 | 67.00±0.39 | 55.98±0.72 | 60.82±0.91 | 62.30 |
| GraphCL | 76.09±1.81 | 67.91±0.96 | 65.44±2.46 | 69.59±0.38 | 61.33±0.65 | 57.12±1.40 | 66.24 |
| KANO | 77.63±6.38 | 92.23±1.33 | 58.64±11.95 | 80.07±2.85 | 68.78±0.84 | 57.45±1.64 | 72.47 |
| ChemBERTa | 63.11±9.96 | 77.18±0.21 | 91.78±3.95 | 70.49±2.23 | 68.70±0.12 | 53.91±1.36 | 70.86 |

Table 11: Fine-tuning and linear probing results of models pretrained on 0.02 M dataset for regression tasks. Note that GraphLog does not provide regression tasks.

Fine-tuning

| Method | ESOL | Lipo | FreeSolv | AVG |
|---|---|---|---|---|
| Grover | 1.344±0.084 | 3.126±0.510 | 0.804±0.019 | 1.758 |
| AttributeMask | 1.463±0.109 | 2.989±0.073 | 0.819±0.036 | 1.757 |
| ContextPred | 1.420±0.091 | 4.382±2.314 | 0.820±0.031 | 2.208 |
| EdgePred | 1.445±0.046 | 3.268±0.342 | 0.841±0.026 | 1.851 |
| GraphLog | - - | - - | -- | |
| GraphCL | 1.018±0.114 | 2.280±0.047 | 0.614±0.021 | 1.304 |
| KANO | 0.639±0.110 | 1.562±0.365 | 0.443±0.007 | 0.881 |
| ChemBERTa | 0.420±0.027 | 4.261±0.503 | 0.598±0.022 | 1.760 |

Linear Probing

| Method | ESOL | Lipo | FreeSolv | AVG |
|---|---|---|---|---|
| Grover | 1.260±0.160 | 3.347±0.587 | 0.801±0.060 | 2.023 |
| AttributeMask | 1.587±0.024 | 3.065±0.020 | 1.090±0.009 | 1.914 |
| ContextPred | 1.989±0.006 | 4.063±0.056 | 1.089±0.006 | 2.380 |
| EdgePred | 2.143±0.007 | 4.048±0.020 | 1.109±0.003 | 2.434 |
| GraphLog | -- | - - | - - | - |
| GraphCL | 1.663±0.051 | 3.353±0.022 | 1.053±0.011 | 1.803 |
| KANO | 0.874±0.050 | 3.196±1.101 | 0.832±0.077 | 1.634 |
| ChemBERTa | 0.420±0.027 | 4.261±0.503 | 0.598±0.022 | 1.760 |

Table 12: Fine-tuning and linear probing results of models pretrained on 0.25 M dataset for regression tasks. Note that GraphLog does not provide regression tasks.

| Fine-tuning | | | | |
|---|---|---|---|---|
| Method | ESOL | Lipo | FreeSolv | AVG |
| Grover | $2.298_{0.255}$ | $3.597_{0.779}$ | $1.054_{0.021}$ | 0.046 |
| AttributeMask | $1.236_{0.066}$ | $2.576_{0.222}$ | $0.801_{0.036}$ | 0.012 |
| ContextPred | $1.212_{0.009}$ | $3.067_{0.257}$ | $0.816_{0.031}$ | 0.009 |
| EdgePred | $1.333_{0.065}$ | $3.102_{0.227}$ | $0.873_{0.026}$ | 0.018 |
| GraphLog | - - | – - | - - | - |
| GraphCL | $1.454_{0.010}$ | $2.978_{0.070}$ | $0.852_{0.019}$ | 0.007 |
| KANO | $0.599_{0.074}$ | $1.442_{0.142}$ | $0.454_{0.007}$ | 0.008 |
| ChemBERTa | $0.394_{0.017}$ | $3.559_{0.147}$ | $0.796_{0.022}$ | 0.035 |
| Linear Probing | | | | |
| Method | ESOL | Lipo | FreeSolv | AVG |
| Grover | $1.430_{0.093}$ | $3.787_{0.170}$ | $0.988_{0.013}$ | 2.068 |
| AttributeMask | $1.921_{0.041}$ | $3.368_{0.008}$ | $1.070_{0.004}$ | 2.120 |
| ContextPred | $1.815_{0.029}$ | $4.041_{0.056}$ | $1.078_{0.014}$ | 2.311 |
| EdgePred | $2.266_{0.019}$ | $4.256_{0.041}$ | $1.111_{0.000}$ | 2.544 |
| GraphLog | - - | - - | - - | - |
| GraphCL | $2.346_{0.253}$ | $3.839_{0.740}$ | $1.077_{0.065}$ | 2.421 |
| KANO | $0.752_{0.128}$ | $2.157_{0.232}$ | $0.755_{0.069}$ | 1.221 |
| ChemBERTa | $0.394_{0.017}$ | $3.559_{0.147}$ | $0.796_{0.035}$ | 1.583 |

Table 13: Fine-tuning and linear probing results of models pretrained on 0.5 M dataset for regression tasks. Note that GraphLog does not provide regression tasks.

| Fine-tuning | | | | |
|---|---|---|---|---|
| Method | ESOL | Lipo | FreeSolv | AVG |
| Grover | $0.951_{0.143}$ | $3.028_{0.613}$ | $0.585_{0.035}$ | 1.521 |
| AttributeMask | $1.269_{0.017}$ | $2.559_{0.068}$ | $0.804_{0.016}$ | 1.544 |
| ContextPred | $1.289_{0.048}$ | $2.926_{0.239}$ | $0.832_{0.013}$ | 1.683 |
| EdgePred | $1.420_{0.051}$ | $2.795_{0.149}$ | $0.796_{0.011}$ | 1.670 |
| GraphLog | - - | - - | - - | - |
| GraphCL | $1.280_{0.040}$ | $4.996_{0.886}$ | $0.845_{0.021}$ | 2.374 |
| KANO | $0.621_{0.105}$ | $1.416_{0.261}$ | $0.442_{0.019}$ | 0.826 |
| ChemBERTa | $0.397_{0.031}$ | $3.935_{0.218}$ | $0.714_{0.008}$ | 1.682 |
| Linear Probing | | | | |
| Method | ESOL | Lipo | FreeSolv | AVG |
| Grover | $1.127_{0.202}$ | $3.391_{0.704}$ | $0.745_{0.060}$ | 1.754 |
| AttributeMask | $1.864_{0.009}$ | $3.284_{0.049}$ | $1.072_{0.008}$ | 2.073 |
| ContextPred | $1.736_{0.041}$ | $3.762_{0.076}$ | $1.054_{0.007}$ | 2.184 |
| EdgePred | $1.984_{0.018}$ | $4.056_{0.120}$ | $1.028_{0.004}$ | 2.356 |
| GraphLog | - - | - - | - - | - |
| GraphCL | $1.955_{0.054}$ | $4.758_{0.251}$ | $0.968_{0.017}$ | 2.560 |
| KANO | $0.754_{0.157}$ | $2.387_{0.384}$ | $0.764_{0.079}$ | 1.302 |
| ChemBERTa | $0.397_{0.031}$ | $3.935_{0.218}$ | $0.714_{0.008}$ | 1.682 |

Table 14: Fine-tuning and linear probing results of models pretrained on 1.0 M dataset for regression tasks. Note that GraphLog does not provide regression tasks.

| Fine-tuning | | | | |
|---|---|---|---|---|
| Method | ESOL | Lipo | FreeSolv | AVG |
| Grover | $0.979_{0.183}$ | $2.768_{0.538}$ | $0.618_{0.012}$ | 1.455 |
| AttributeMask | $1.304_{0.006}$ | $2.690_{0.148}$ | $0.796_{0.022}$ | 1.597 |
| ContextPred | $1.272_{0.021}$ | $2.913_{0.145}$ | $0.847_{0.019}$ | 1.677 |
| EdgePred | $1.472_{0.086}$ | $2.366_{0.273}$ | $0.840_{0.003}$ | 1.559 |
| GraphLog | - - | - - | - - | - |
| GraphCL | $1.351_{0.037}$ | $3.387_{1.075}$ | $0.841_{0.014}$ | 1.860 |
| KANO | $0.627_{0.087}$ | $1.389_{0.192}$ | $0.447_{0.005}$ | 0.821 |
| ChemBERTa | $0.434_{0.004}$ | $3.966_{0.158}$ | $0.748_{0.044}$ | 1.716 |
| Linear Probing | | | | |
| Method | ESOL | Lipo | FreeSolv | AVG |
| Grover | $1.091_{0.265}$ | $3.104_{0.496}$ | $0.741_{0.057}$ | 1.645 |
| AttributeMask | $1.902_{0.018}$ | $3.366_{0.106}$ | $1.068_{0.004}$ | 2.112 |
| ContextPred | $1.706_{0.018}$ | $3.881_{0.126}$ | $1.056_{0.003}$ | 2.214 |
| EdgePred | $1.985_{0.018}$ | $4.056_{0.120}$ | $1.030_{0.007}$ | 2.357 |
| GraphLog | - - | - - | - - | - |
| GraphCL | $1.589_{0.040}$ | $4.583_{0.248}$ | $1.012_{0.016}$ | 2.395 |
| KANO | $0.782_{0.131}$ | $2.277_{0.414}$ | $0.762_{0.074}$ | 1.273 |
| ChemBERTa | $0.434_{0.004}$ | $3.966_{0.158}$ | $0.748_{0.044}$ | 1.716 |

Table 15: Fine-tuning and linear probing results of models pretrained on 1.5 M dataset for regression tasks. Note that GraphLog does not provide regression tasks.

| Fine-tuning | | | | |
|---|---|---|---|---|
| Method | ESOL | Lipo | FreeSolv | AVG |
| Grover | $1.334_{0.064}$ | $3.468_{0.555}$ | $0.801_{0.005}$ | 1.867 |
| AttributeMask | $1.338_{0.101}$ | $3.189_{0.284}$ | $0.827_{0.007}$ | 1.785 |
| ContextPred | $1.484_{0.053}$ | $3.448_{0.543}$ | $0.833_{0.029}$ | 1.922 |
| EdgePred | $1.472_{0.068}$ | $3.363_{0.361}$ | $0.851_{0.014}$ | 1.895 |
| GraphLog | - - | - - | - - | - |
| GraphCL | $0.957_{0.126}$ | $2.932_{0.517}$ | $0.620_{0.002}$ | 1.503 |
| KANO | $0.600_{0.067}$ | $1.455_{0.047}$ | $0.423_{0.008}$ | 0.826 |
| ChemBERTa | $0.403_{0.019}$ | $3.683_{0.269}$ | $0.616_{0.009}$ | 1.567 |
| Linear Probing | | | | |
| Method | ESOL | Lipo | FreeSolv | AVG |
| Grover | $1.635_{0.034}$ | $3.255_{0.120}$ | $1.087_{0.120}$ | 1.992 |
| AttributeMask | $1.746_{0.087}$ | $3.298_{0.085}$ | $1.082_{0.085}$ | 2.042 |
| ContextPred | $2.014_{0.011}$ | $3.632_{0.133}$ | $1.091_{0.133}$ | 2.246 |
| EdgePred | $2.188_{0.018}$ | $4.337_{0.043}$ | $1.107_{0.043}$ | 2.544 |
| GraphLog | - - | - - | - - | - |
| GraphCL | $1.105_{0.265}$ | $3.414_{0.752}$ | $0.756_{0.752}$ | 1.758 |
| KANO | $0.733_{0.157}$ | $2.122_{0.490}$ | $0.746_{0.490}$ | 1.201 |
| ChemBERTa | $0.403_{0.019}$ | $3.683_{0.269}$ | $0.616_{0.269}$ | 1.567 |

Table 16: Fine-tuning and linear probing results of models pretrained on 2.0 M dataset for regression tasks. Note that GraphLog does not provide regression tasks.

**Fine-tuning**

| Method | ESOL | Lipo | FreeSolv | AVG |
|---|---|---|---|---|
| Grover | $1.369_{0.042}$ | $2.489_{0.138}$ | $0.825_{0.019}$ | 1.561 |
| AttributeMask | $1.234_{0.024}$ | $2.645_{0.121}$ | $0.791_{0.020}$ | 1.557 |
| ContextPred | $1.330_{0.030}$ | $2.854_{0.220}$ | $0.814_{0.015}$ | 1.666 |
| EdgePred | $1.442_{0.049}$ | $2.926_{0.081}$ | $0.821_{0.012}$ | 1.730 |
| GraphLog | - - | - - | - - | - |
| GraphCL | $1.008_{0.178}$ | $2.946_{0.755}$ | $0.581_{0.031}$ | 1.512 |
| KANO | $0.602_{0.103}$ | $1.512_{0.153}$ | $0.431_{0.006}$ | 0.848 |
| ChemBERTa | $0.368_{0.033}$ | $3.773_{0.281}$ | $0.604_{0.019}$ | 1.582 |

**Linear Probing**

| Method | ESOL | Lipo | FreeSolv | AVG |
|---|---|---|---|---|
| Grover | $1.565_{0.088}$ | $3.317_{0.344}$ | $0.992_{0.020}$ | 1.958 |
| AttributeMask | $1.872_{0.018}$ | $3.522_{0.071}$ | $1.062_{0.014}$ | 2.152 |
| ContextPred | $1.760_{0.033}$ | $3.907_{0.043}$ | $1.060_{0.005}$ | 2.242 |
| EdgePred | $2.006_{0.032}$ | $4.388_{0.057}$ | $1.078_{0.018}$ | 2.491 |
| GraphLog | - - | - - | - - | - |
| GraphCL | $1.097_{0.242}$ | $3.209_{0.752}$ | $0.761_{0.065}$ | 1.689 |
| KANO | $0.712_{0.160}$ | $2.089_{0.466}$ | $0.753_{0.073}$ | 1.184 |
| ChemBERTa | $0.368_{0.033}$ | $3.773_{0.281}$ | $0.604_{0.019}$ | 1.582 |

Table 17: Prediction performance on six downstream tasks and the overall average (across 3 repeats) using scaffold splitting, reported in terms of ROC-AUC ($\uparrow$) as mean and standard deviation in %. The setting is the same as in the main experiments, except that the hidden dimension is increased to 1200.

**(A) Fine-tuning**

| Method | BACE | BBBP | ClinTox | Tox21 | ToxCast | SIDER | AVG |
|---|---|---|---|---|---|---|---|
| GROVER | $84.82_{\pm3.16}$ | $92.96_{\pm1.44}$ | $83.22_{\pm2.19}$ | $85.02_{\pm0.48}$ | $71.69_{\pm0.96}$ | $61.85_{\pm1.08}$ | 79.9 |
| AttributeMask | $80.40_{\pm2.31}$ | $69.54_{\pm0.57}$ | $80.64_{\pm4.64}$ | $74.49_{\pm0.58}$ | $63.68_{\pm0.36}$ | $57.80_{\pm1.85}$ | 71.0 |
| ContextPred | $76.91_{\pm1.49}$ | $66.79_{\pm1.82}$ | $69.98_{\pm4.80}$ | $74.79_{\pm0.86}$ | $64.80_{\pm0.65}$ | $62.28_{\pm0.89}$ | 69.2 |
| EdgePred | $67.42_{\pm4.27}$ | $67.35_{\pm1.11}$ | $58.66_{\pm6.28}$ | $73.07_{\pm0.89}$ | $61.70_{\pm0.91}$ | $57.38_{\pm1.74}$ | 64.2 |
| GraphLoG | $82.69_{\pm0.86}$ | $65.57_{\pm1.89}$ | $67.24_{\pm4.63}$ | $72.26_{\pm1.50}$ | $61.98_{\pm0.88}$ | $59.09_{\pm1.66}$ | 68.6 |
| GraphCL | $78.06_{\pm3.07}$ | $63.86_{\pm4.41}$ | $64.64_{\pm7.35}$ | $74.05_{\pm2.23}$ | $63.36_{\pm0.29}$ | $59.09_{\pm1.66}$ | 67.1 |
| KANO | $84.20_{\pm1.34}$ | $93.05_{\pm2.27}$ | $84.66_{\pm6.12}$ | $83.54_{\pm2.39}$ | $72.36_{\pm1.06}$ | $59.93_{\pm2.82}$ | 79.6 |
| ChemBERTa | $80.35_{\pm1.28}$ | $74.18_{\pm1.10}$ | $73.06_{\pm12.2}$ | $71.21_{\pm1.76}$ | $67.55_{\pm1.60}$ | $56.15_{\pm2.21}$ | 70.4 |

**(B) Linear probing**

| Method | BACE | BBBP | ClinTox | Tox21 | ToxCast | SIDER | AVG |
|---|---|---|---|---|---|---|---|
| GROVER | $84.17_{\pm3.43}$ | $92.01_{\pm4.50}$ | $78.27_{\pm9.49}$ | $82.82_{\pm2.18}$ | $67.93_{\pm0.93}$ | $62.65_{\pm3.16}$ | 77.97 |
| AttributeMask | $64.55_{\pm1.13}$ | $62.97_{\pm0.96}$ | $55.51_{\pm2.02}$ | $67.66_{\pm0.55}$ | $57.56_{\pm0.13}$ | $56.20_{\pm0.90}$ | 60.74 |
| ContextPred | $74.94_{\pm1.03}$ | $64.36_{\pm0.46}$ | $53.31_{\pm1.79}$ | $68.36_{\pm0.51}$ | $58.75_{\pm0.66}$ | $59.35_{\pm0.65}$ | 63.18 |
| EdgePred | $53.42_{\pm7.26}$ | $53.75_{\pm2.90}$ | $49.36_{\pm1.10}$ | $50.37_{\pm0.34}$ | $50.94_{\pm0.25}$ | $50.39_{\pm1.09}$ | 51.37 |
| GraphLoG | $72.87_{\pm1.03}$ | $59.65_{\pm0.84}$ | $60.14_{\pm0.71}$ | $68.36_{\pm0.07}$ | $57.56_{\pm0.58}$ | $57.72_{\pm0.75}$ | 62.72 |
| GraphCL | $70.94_{\pm2.02}$ | $61.79_{\pm0.65}$ | $61.44_{\pm1.60}$ | $70.93_{\pm0.74}$ | $59.88_{\pm0.37}$ | $59.70_{\pm0.77}$ | 64.11 |
| KANO | $82.23_{\pm6.07}$ | $93.28_{\pm2.88}$ | $53.58_{\pm12.9}$ | $82.08_{\pm2.80}$ | $68.91_{\pm1.15}$ | $60.71_{\pm1.97}$ | 73.46 |
| ChemBERTa | $79.54_{\pm0.23}$ | $75.57_{\pm0.48}$ | $13.93_{\pm1.30}$ | $69.41_{\pm0.62}$ | $67.09_{\pm0.80}$ | $52.63_{\pm0.45}$ | 59.69 |

**(C) Random Initialization (Fine-tuning)**

| Method | BACE | BBBP | ClinTox | Tox21 | ToxCast | SIDER | AVG |
|---|---|---|---|---|---|---|---|
| GROVER | $84.50_{\pm4.07}$ | $92.93_{\pm4.40}$ | $83.99_{\pm7.13}$ | $85.05_{\pm1.97}$ | $71.52_{\pm0.95}$ | $62.18_{\pm1.18}$ | 80.03 |
| AttributeMask | $61.15_{\pm2.90}$ | $67.35_{\pm1.19}$ | $53.56_{\pm7.97}$ | $72.46_{\pm0.56}$ | $58.53_{\pm0.42}$ | $54.65_{\pm3.33}$ | 61.28 |
| ContextPred | $72.96_{\pm1.75}$ | $68.14_{\pm2.92}$ | $50.87_{\pm6.78}$ | $72.60_{\pm0.74}$ | $60.21_{\pm1.62}$ | $57.34_{\pm2.68}$ | 63.69 |
| EdgePred | $65.87_{\pm5.00}$ | $67.45_{\pm1.28}$ | $57.00_{\pm13.2}$ | $70.64_{\pm2.20}$ | $59.24_{\pm0.37}$ | $53.31_{\pm4.58}$ | 62.25 |
| GraphLoG | $69.45_{\pm10.8}$ | $65.35_{\pm4.15}$ | $52.61_{\pm9.73}$ | $72.10_{\pm2.05}$ | $58.95_{\pm0.97}$ | $54.27_{\pm5.39}$ | 62.12 |
| GraphCL | $69.57_{\pm6.68}$ | $67.12_{\pm2.27}$ | $49.61_{\pm4.93}$ | $70.91_{\pm1.20}$ | $60.13_{\pm1.58}$ | $55.86_{\pm5.55}$ | 62.20 |
| KANO | $83.35_{\pm0.99}$ | $93.61_{\pm1.52}$ | $88.31_{\pm2.68}$ | $84.00_{\pm2.93}$ | $71.30_{\pm0.98}$ | $60.80_{\pm1.13}$ | 80.23 |
| ChemBERTa | $77.20_{\pm1.89}$ | $66.85_{\pm1.58}$ | $43.69_{\pm14.0}$ | $66.72_{\pm1.36}$ | $67.16_{\pm2.86}$ | $50.96_{\pm1.87}$ | 62.10 |

**(D) Random Initialization (Linear probing)**

| Method | BACE | BBBP | ClinTox | Tox21 | ToxCast | SIDER | AVG |
|---|---|---|---|---|---|---|---|
| GROVER | $83.59_{\pm3.43}$ | $92.14_{\pm4.50}$ | $79.20_{\pm9.49}$ | $83.83_{\pm2.18}$ | $68.07_{\pm0.93}$ | $61.98_{\pm3.16}$ | 81.24 |
| AttributeMask | $67.83_{\pm1.13}$ | $65.29_{\pm0.96}$ | $58.91_{\pm2.02}$ | $67.52_{\pm0.55}$ | $56.49_{\pm0.13}$ | $56.25_{\pm0.90}$ | 62.05 |
| ContextPred | $64.37_{\pm1.03}$ | $66.30_{\pm0.46}$ | $57.49_{\pm1.79}$ | $69.48_{\pm0.51}$ | $59.62_{\pm0.66}$ | $58.02_{\pm0.65}$ | 62.55 |
| EdgePred | $62.20_{\pm7.26}$ | $66.26_{\pm2.90}$ | $57.60_{\pm1.10}$ | $69.24_{\pm0.34}$ | $58.64_{\pm0.25}$ | $57.87_{\pm1.09}$ | 61.97 |
| GraphLoG | $64.89_{\pm1.03}$ | $66.47_{\pm0.84}$ | $59.98_{\pm0.71}$ | $58.26_{\pm0.07}$ | $69.32_{\pm0.58}$ | $59.05_{\pm0.75}$ | 62.99 |
| GraphCL | $62.76_{\pm2.02}$ | $66.35_{\pm0.65}$ | $57.41_{\pm1.60}$ | $69.11_{\pm0.74}$ | $59.63_{\pm0.37}$ | $57.67_{\pm0.77}$ | 62.16 |
| KANO | $82.00_{\pm6.07}$ | $94.43_{\pm2.88}$ | $74.86_{\pm12.9}$ | $85.28_{\pm2.80}$ | $76.66_{\pm1.15}$ | $63.84_{\pm1.97}$ | 79.51 |
| ChemBERTa | $76.72_{\pm0.23}$ | $73.23_{\pm0.48}$ | $42.69_{\pm1.30}$ | $69.01_{\pm0.62}$ | $73.33_{\pm0.80}$ | $57.55_{\pm0.45}$ | 65.42 |

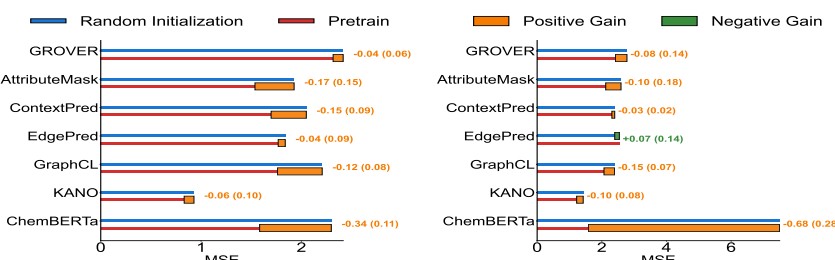

Figure 8: This figure shows the Pretrain Gain of linear probing and fine-tuning when trained on 0.25 M samples. Unlike in classification, a negative value here indicates that the model has benefited from pretraining, while a positive value suggests that it did not gain from pretraining.

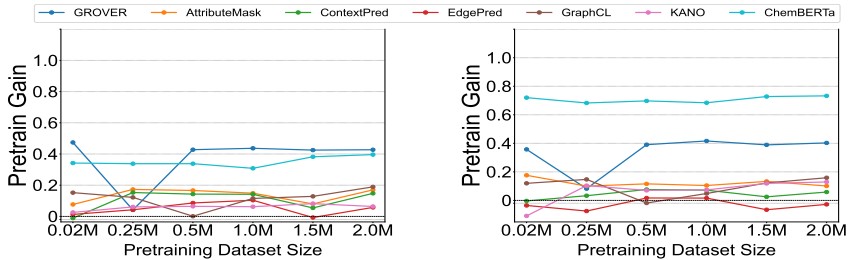

Figure 9: The figure illustrates the Pretrain Gain for fine-tuning and linear probing in regression tasks. A smaller performance gap between the two indicates that linear probing achieves high performance, suggesting that the pretrained representations are highly generalizable. Note that GraphLog is excluded from this analysis, as it does not provide code support for regression tasks. Originally, since this is a regression task, negative values for linear probing would indicate better performance. However, for intuitive interpretation, the signs have been reversed — meaning that higher values indicate greater Pretrain Gain.

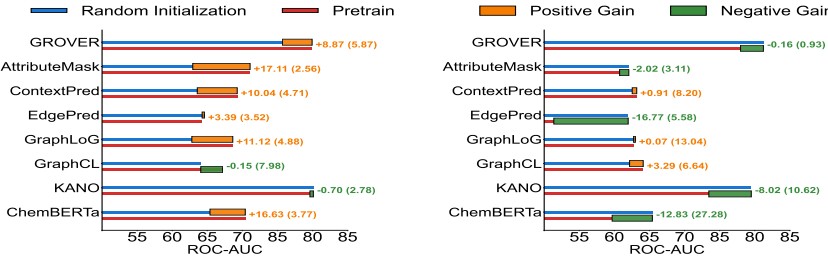

Figure 10: This figure shows the Pretrain Gain of linear probing and fine-tuning when trained on 0.25M samples, with the hidden dimension increased to 1200.

