# OpenReview forum: "Fine-tuning is Not Enough: Rethinking Evaluation in Molecular Self-Supervised Learning"
_ICLR.cc/2026/Conference — ICLR 2026 Conference Withdrawn Submission_

### Official Review · Reviewer_94XP · 2025-10-29

**Soundness:** 3
**Presentation:** 2
**Contribution:** 2
**Rating:** 4
**Confidence:** 4

**Summary:**

This paper revisits evaluation methodologies in molecular self-supervised learning. It argues that the prevalent fine-tuning performance metric fails to fully capture the generalization and representation quality of pretrained models. To address this, the authors propose a multi-perspective evaluation framework consisting of:
- Linear probing to test frozen encoder representations,
- Pretrain Gain, measuring improvements over random initialization,
- Parameter Shift to quantify forgetting during fine-tuning, and
- Scalability analysis across dataset sizes.

Experiments on eight representative molecular SSL methods (GROVER, GraphCL, KANO, ChemBERTa, etc.) reveal surprising findings: many models show low or even negative Pretrain Gain under linear probing, substantial parameter shifts for GNNs, and negligible scalability. The study provides a diagnostic reassessment of how molecular SSL should be evaluated.

**Strengths:**

1. The paper addresses a key methodological weakness in current molecular SSL literature, the overreliance on fine-tuning metrics, and offers a concrete alternative evaluation paradigm.
2. Comprehensive experimental design: The authors control for non-pretraining factors (architecture, datasets, heads, dimensions), enabling fair cross-model comparison. This is rare in molecular ML.
3. Empirical insights: The finding that some SSL models yield negative Pretrain Gain is striking and valuable for the community.

**Weaknesses:**

1. Insufficient comparative experiments. Lack of comparative results of HIV and MUV datasets; If multiple evaluation indicators are related to parameters, please add the analysis of the parameters related to the prediction head.
2. Lack of theoretical foundation. Pretrain gain/parameter shift, while intuitive, lacks theoretical foundation. On the other hand, fine-tuning is to better adapt the model to new tasks, which cannot account for forgetfulness and is not conducive to downstream tasks.
3. The comparison of Figure 2 is unreasonable. Random initialization does not learn about pre-training, and cannot explain the difference between linear probing and FT.
4. Scalability section is shallow. Although dataset sizes vary, the study does not explore larger orders of magnitude, limiting conclusions about true scaling laws.

**Questions:**

1. Figure 2 How does the author explain the phenomenon of overemphasizing negative returns, but in fact FT helps to obtain positive gains?
2. How sensitive are your results to the downstream head architecture (e.g., 1-layer vs. 2-layer MLP)?
3. Can I visualize the difference in weights before and after fine-tuning (Eq.2)?
4. Could you add results with larger pretraining datasets (e.g., >10M) or pretrained models from prior work to further validate the scalability?
5. Have you considered evaluating out-of-domain generalization as part of the framework?
6. See weakness.

---

### Official Review · Reviewer_VNaQ · 2025-10-31

**Soundness:** 2
**Presentation:** 2
**Contribution:** 2
**Rating:** 2
**Confidence:** 4

**Summary:**

This paper points out the limitation in traditional evaluation of molecular pretrained models primarily based on fine-tuning performance with inconsistent setup, and thus proposes a new evaluation paradigm for molecular pretrained models with more unified setting and evaluation metrics across different perspectives to better understand and isolate the effect from pretraining.

**Strengths:**

- This paper indeed points out the previous limitation and difficulty to obtain a more fair and unified performance comparison over various molecular pretrained models
- Through their designed benchmark evaluation across different perspectives, there are several insights regarding different architecture of the pre-training models and they include a study on data scaling effect

**Weaknesses:**

- As a benchmark paper that particularly aiming for molecular pretrained models, the baselines used for evaluation are rather outdated, lack of latest and more powerful molecular pretraining models, like [1, 3, 4]
- The datasets for pretraining and evaluation are also small and outdated, Zinc is only 2M, we can include larger datasets for pretraining, for instance the datasets include in [2]. Also, the evaluation datasets only contain classification tasks without any regression tasks, like some basic ones: esol, lipo, ...
- Although the idea that we want to evaluate over not only fine-tuning performance is valid and reasonable, but the metrics design might not be that precise and at a high level, for instance, the forgetting is reflected by the distance in parameters. Then, should we also have a more comprehensive definition and evaluation of the forgetting of the knowledge instead of referring only to the parameters.
- The paper presentation needs some refinement, especially the quality of the plots included are not that professional.

[1]. Xia, Jun, et al. "Mole-bert: Rethinking pre-training graph neural networks for molecules." (2023).

[2]. Beaini, Dominique, et al. "Towards foundational models for molecular learning on large-scale multi-task datasets." arXiv preprint arXiv:2310.04292 (2023).

[3]. Luo, Yizhen, et al. "Molfm: A multimodal molecular foundation model." arXiv preprint arXiv:2307.09484 (2023).

[4]. Ji, Xiaohong, et al. "Uni-mol2: Exploring molecular pretraining model at scale." arXiv preprint arXiv:2406.14969 (2024).

**Questions:**

- Do you expect the evaluation of current framework can be extended to 3D pretrained models and the pretraining including LLM in the process and generalize to future molecular foundation models
- others please see weakness

---

### Official Review · Reviewer_eMxA · 2025-11-01

**Soundness:** 1
**Presentation:** 2
**Contribution:** 3
**Rating:** 2
**Confidence:** 4

**Summary:**

This paper presents a multi-perspective evaluation framework for molecular self-supervised learning (SSL) that extends beyond the traditional fine-tuning-only evaluation common in this domain. The authors propose four evaluation strategies: (1) linear probing to assess learned representation quality, (2) pretrain gain to quantify benefits of pretraining versus random initialization, (3) parameter shift analysis to measure forgetting during fine-tuning, and (4) scalability experiments with varying pretraining dataset sizes. Under a unified experimental setup controlling for hidden dimensions and downstream architecture, the paper evaluates eight molecular SSL methods (GROVER, AttributeMask, ContextPred, EdgePred, GraphLoG, GraphCL, KANO, ChemBERTa) on six downstream molecular property prediction tasks. Key findings include: several models exhibit low or negative pretrain gain in linear probing, GNN-based models experience substantial parameter shifts during fine-tuning, and most models show negligible benefits from larger pretraining datasets.

**Strengths:**

- The empirical work is extensive and demonstrates significant effort:
    - Training and evaluating eight SSL methods across multiple downstream tasks, dataset scales, and multiple repetitions represents a substantial computational and organizational undertaking.
    - The attempt to standardize experimental conditions (hidden dimensions, prediction heads, datasets) across diverse methods identifies (and aims to address) a real gap in prior work where each method was evaluated under different conditions.
- The paper makes a valuable contribution by questioning the sole reliance on fine-tuning performance as an evaluation metric for molecular SSL:
    - The proposed multi-perspective evaluation framework, particularly the pretrain gain metric, provides a more nuanced view of pretraining effectiveness.
    - The unified experimental setup represents a commendable effort to enable fair comparisons across methods.
- The paper tackles an important problem in molecular SSL evaluation:
    - The finding that high fine-tuning performance does not guarantee high-quality pretrained representations (e.g., KANO's negligible pretrain gain despite top fine-tuning performance) is a critical insight.
    - This could influence future research directions in how molecular SSL methods are evaluated.

**Weaknesses:**

- There are some serious issues with the experimental design that limit the conclusions that can be drawn:
    - The authors confound architecture and pretraining strategy:
        - Architecture (GNN vs. Transformer) and pretraining strategy (e.g., masking, contrastive learning) are two independent axes that should be evaluated separately.
        - A proper experimental design would either: (1) evaluate all combinations in a grid (each pretraining strategy on each architecture), or (2) hold one axis constant while varying the other.
        - Instead, the paper confounds these factors: GNN-based models use molecular graphs while Transformer-based models use SMILES strings, and each model uses its own specific pretraining strategy. Hybrid approaches further complicate the analysis rather than clarifying it.
        - This makes it impossible to determine whether performance differences arise from the architecture choice, the pretraining strategy, the input modality, or some interaction between them. This severely limits the ability to draw conclusions about the effectiveness of pretraining strategies themselves.
    - The authors do not control for model capacity in a principled manner:
        - Fixing hidden dimension to 300 across all models does not ensure fair comparison. Hidden dimension means different things for different architectures (e.g., Transformer attention heads vs. GNN message passing dimension).
        - The appropriate comparison would control for total parameter count, not hidden dimension.
    - Critical regularization details (weight decay, dropout rates, etc.) are not discussed, and it's unclear whether these were controlled across models or left at their original paper-specific settings:
        - These have major impact on fine-tuning behavior and the parameter shift metric.
        - This omission makes it difficult to interpret the results or reproduce the experiments.
- The parameter shift metric (Eq. 2) is seriously flawed:
    - It sums L2 distances across all parameters, making it fundamentally incomparable across models with different architectures.
    - Models have different parameter counts, yet the metric is an absolute sum.
    - Parameters have different scales across architectures (e.g., normalization layers vs. weight matrices).
    - Weight decay and other regularization techniques significantly affect this metric, yet are not discussed.
    - Reparameterizations that produce identical functional outputs yield different shift values. For example, consider a simple model $f(x, W) = \text{LN}(xW)$. Both $f(x, W)$ and $f(x, cW)$ (where $c$ is a constant) produce the same output, but have different parameter shift values.
    - The correlation shown in Figure 7 between parameter shift and performance gap does not validate the metric's utility for cross-model comparison.
    - A better approach: use functional similarity metrics such as CKA or CCA on hold-out embeddings from pretrained vs. fine-tuned models, which are architecture-agnostic and measure what matters---representation similarity.
- The regression results have significant presentation issues:
    - Tables 11-16 appear to average MSE across datasets with different target scales, which is meaningless as MSE values for different regression tasks are not comparable.
    - Further, because the scales of the targets and MSEs are different, it's much harder to interpret all the other results for regression (e.g., pretrain gain figure).
    - Finally, there seems to be a reversal of linear probing trends for regression versus classification. This is interesting but not discussed adequately (please see my question about this below).
- While linear probing is a reasonable metric to include in such a study, its utility for evaluating representation quality in deep pretrained models is questionable:
    - Linear probing evaluates whether frozen representations are directly applicable to downstream tasks with minimal adaptation (i.e., only a linear or shallow classifier). However, in modern deep learning (vision, NLP, etc.), adapter-based methods and LoRA consistently extract substantially more performance from the same pretrained models while maintaining comparable parameter efficiency. This suggests that the ability to perform well under linear probing is not strongly indicative of the overall quality or usefulness of learned representations---rather, it reflects one specific (and arguably limited) notion of representation utility.
    - An interesting alternative direction would have been to evaluate these models using parameter-efficient fine-tuning methods like LoRA alongside (or instead of) linear probing, as this might provide more reliable insights into representation quality.
- The model scaling experiments are underdeveloped:
    - The main paper focuses primarily on data scaling but appears to also have conducted model scaling experiments with the hidden dimension 1200 results (Table 17).
    - If extensive model scaling experiments were done, it would be good to present them in a similar manner to the data scaling experiments: Create a visualization showing performance as a function of model size at fixed data size.
    - If the full grid of model sizes and data sizes have been explored, then some visualization of the full scaling surface would be valuable (perhaps some sort of Pareto front analysis).
    - The existing model scaling results should be better integrated into the main discussion.
- Many figures have poor aspect ratios that impede readability (Figures 2, 4, 5, 6, 8, 9, 10).

**Questions:**

1. What are the weight decay settings for each model? How sensitive are your conclusions to these regularization choices?
2. Looking at the regression results, it seems like the trends that were observed for linear probing (that about half the methods showed negative pretrain gain) are now gone for regression tasks (i.e., fine-tuning always helps). Provided that this was a major finding in the classification experiments, can you discuss why the regression results differ? Is this due to differences in task nature, dataset characteristics, or something else?
    - The case of ChemBERTa is a particularly interesting one to investigate further, as the linear probing pretrain gain is massive for regression, but fairly negligible for classification. What do you think explains this discrepancy?

**Minor Comments and Nits**

- Line 152: "GNNs are used to extract graph structure" appears to be an incomplete sentence.
- Line 213: "fine-tuning" should be capitalized.

---

### Official Review · Reviewer_SvRS · 2025-11-01

**Soundness:** 3
**Presentation:** 3
**Contribution:** 3
**Rating:** 4
**Confidence:** 4

**Summary:**

The paper introduces an evaluation framework for SSL methods in the molecular domain. In this work, a scaffold split is employed as a standard approach to evaluate model performance on unknown scaffolds, thereby testing the methods' ability to generalize. The primary objective of this research is to compare multiple molecular SSL methods by measuring the performance gain achieved when the entire model is fine-tuned versus when only a linear prediction head is trained. Additionally, knowledge forgetting is quantified in the fine-tuned models by calculating the difference in weights before and after fine-tuning. Finally, the study examines the relationship between the size of the pre-training dataset and model performance. This research leads to several noteworthy conclusions. For example, transformer models are less likely to forget knowledge learned during pre-training, and some models produce representations that are less useful for the target task than embeddings generated by a randomly initialized model.

**Strengths:**

- The study includes different families of SSL models, which leads to interesting conclusions about how different architectures and pretraining techniques behave.
- All experiments were repeated three times, and standard deviation values are reported, making the evaluation more robust.
- A scaffold split is adopted, which is a standard for evaluating model generalization in the molecular domain.
- The evaluation offers various perspectives on pretrained models: performance gain, knowledge forgetting, and scalability.
- The conclusions drawn from this study may be useful for researchers working on the molecular SSL methods or readers who want to find a useful pretraining scheme.
- The paper is well-written, with a clear evaluation objective.

**Weaknesses:**

- The techniques used for evaluating SSL approaches in this work are not new, and many of them were adapted from the papers introducing SSL methods or from other domains (for example, linear probing was used in SimCLR, and the difference of weights was evaluated here [1]).
- Some figures are of low quality. For example, Figure 2 appears to have been scaled down in height.
- There are numerous references to tables and figures in the appendix, and the main text does not make sense without consulting these tables and figures. The main text of the paper should be self-contained, with the appendix serving only to explain experimental details and to present additional experiments.
- The code is not available, which makes it impossible for other researchers to use this framework (scaffold splits should be pre-defined for better reproducibility, and all the metrics should be clearly defined).
- (minor) Section 3.3 begins with a small letter.

[1] Li, D., & Zhang, H. (2021). Improved regularization and robustness for fine-tuning in neural networks. Advances in Neural Information Processing Systems, 34, 27249-27262.

**Questions:**

1. Have you verified whether weight drift actually causes forgetting? For example, you could verify whether these models can still perform the original pretraining task on a validation set, achieving a loss comparable to that of the original pre-trained weights.
2. Do you think that the flat pretrain gain plots (Figure 4) may be caused by the small number of model parameters, or are they the inherent inability of graph models to learn meaningful information from these pre-training tasks?

---

### Note · Authors · 2025-11-20

I have read and agree with the venue's withdrawal policy on behalf of myself and my co-authors.